# IS FREE SELF-ALIGNMENT POSSIBLE?

## ABSTRACT

Aligning pretrained language models (LMs) is a complex and resource-intensive process, often requiring access to large amounts of ground-truth preference data and substantial compute. Are these costs necessary? That is, *is it possible to align using only inherent model knowledge and without additional training?* We tackle this challenge with ALIGNEZ, a novel approach that uses (1) self-generated preference data and (2) representation editing to provide nearly cost-free alignment. During inference, ALIGNEZ modifies LM representations to reduce undesirable and boost desirable components using subspaces identified via self-generated preference pairs. Our experiments reveal that this nearly cost-free procedure significantly narrows the gap between base pretrained and tuned models by an average of 29.1%, observed across five datasets and two model architectures. Additionally, we explore the potential of using ALIGNEZ as a means of *expediting* more expensive alignment procedures. Our experiments show that ALIGNEZ improves DPO models tuned only using a small subset of ground-truth preference data.

## 1 INTRODUCTION

Large language model (LMs) alignment involves the use of complex and expensive pipelines (Schulman et al., 2017; Ouyang et al., 2022; Rafailov et al., 2024). Usually at least two critical components are needed: (1) collecting human preference data, and (2) modifying pretrained model weights to better align with these preferences. Some pipelines involve more complexity (e.g., RLHF trains a reward model on the human preference data and uses it for PPO-based model optimization). Such approaches face substantial scalability challenges: collecting human preference data is costly and time-intensive, and as model sizes increase, the computational requirements for fine-tuning are likely to become prohibitive.

A prospective way to bypass the need for human preference data is to exploit knowledge ***already contained*** in the pretrained model weights. This idea is motivated by evidence suggesting that alignment merely reveals knowledge and capabilities acquired during pretraining (Zhou et al., 2024; Lin et al., 2023). This notion has led to a growing body of literature achieving impressive results using signal contained in pretrained models for fine-tuning (Fränken et al., 2024; Wang et al., 2022; Sun et al., 2023; 2024), largely or totally sidestepping human annotation.

Next, to achieve free alignment, we must additionally obviate the need for fine-tuning. Instead, we propose to replace it with a form of ***representation editing*** that does not require computing gradients or even optimizing a proxy loss at all. Existing representation editing approaches (Zou et al.; Wu et al., 2024; Li et al., 2024) rely on access to ground truth data, which does not account for the unique challenges of using only signals from pretrained models. These signals are often noisier and more limited compared to human-annotated data (Bender et al., 2021; Bommasani et al., 2021; Kenton et al., 2021; Tamkin et al., 2021), necessitating a more tailored approach.

This work puts together these two pieces to *explore the feasibility of free self-alignment*. We align pretrained LMs to human preferences using only the knowledge from the model itself, without additional training or fine-tuning. Procedures that can accomplish these two goals are motivated by an area of need—performing fast alignment repeatedly at-scale for on-the-fly model personalization. In such scenarios, we lack the time and training resources to acquire manually-annotated data and wait for models using RLHF or DPO techniques. Indeed, with limited time—and thus being limited to training on few samples—DPO will fail to achieve any meaningful level of alignment, while

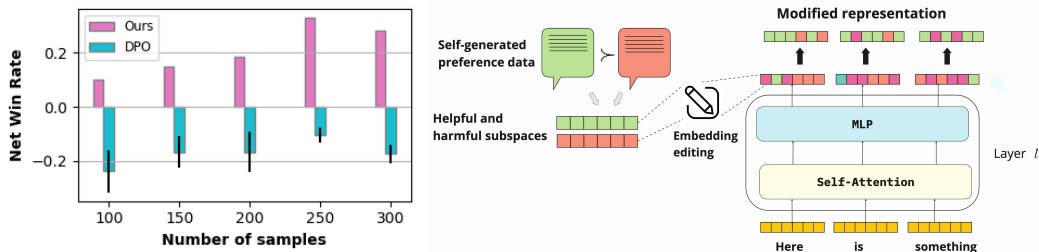

Figure 1: **Left**: Training with DPO (blue) in time-constrained scenarios permits using only a few samples and produces poor alignement even as the number of samples increases (x-axis). In contrast, ALIGNEZ (pink) achieves alignment gains even with limited time, as it is *training free*. **Right**: ALIGNEZ identifies helpful and harmful subspaces for alignment (left)—using only self-generated data. These enable modifying representations during inference (right).

techniques that can accomplish free self-alignment will easily outperform it. We show an example of such scenarios in Fig. 1.

We introduce ALIGNEZ, a novel approach designed for this setting. Using the pretrained model's own generated preference pairs, ALIGNEZ identifies the subspaces within the model's embedding spaces that correspond to helpful and non-helpful responses. During inference, we surgically modify the model's embeddings by boosting the components from the helpful subspaces and neutralizing those from the non-helpful ones.

With this nearly cost-free procedure, we effectively narrow the performance gap between pretrained and aligned models by 29.1% across two model architectures and five datasets. Additionally, we explore the potential of ALIGNEZ to expedite more expensive alignment processes. Our experimental results demonstrate that ALIGNEZ improves upon models trained using DPO (Rafailov et al., 2024) with only a small subset of ground-truth preference data. In summary, our contributions include:

1. We introduce ALIGNEZ, a nearly cost-free approach that leverages preference data generated by the pretrained LM to modify its embeddings, aligning outputs to human preferences.

2. Our experiments show that ALIGNEZ significantly narrows the gap between the base model and its counterparts aligned with traditional expensive methods by 29.1% across two model architectures and five datasets.

3. We demonstrate that ALIGNEZ can *expedite* more expensive methods like DPO by improving models trained with DPO using only a small subset of ground truth preference data. Remarkably, ALIGNEZ boosts the performance of a model trained on only 1% of the data to match that of one trained on 25%.

> Our work suggests that models may be effectively steered, without additional training or supervision. Using the strategies we have developed, ***we envision the possibility of new techniques that go far beyond alignment as it exists today***, tackling such areas as fine-grained and real-time personalization, that are currently beyond the reach of existing methods.

## 2  ALIGNEZ: (ALMOST) FREE ALIGNMENT OF LANGUAGE MODELS

We are ready to describe the ALIGNEZ algorithm. First, we query a base pretrained LM to generate its own preference data. Our intuition is that, while noisy, base models have learned, from pretraining data, sufficient signal to aid in alignment. Using this self-generated data, the identify the subspaces in the LM's embedding spaces that correspond to helpful and harmful directions for alignment. During inference, we modify the LM embeddings using these identified subspaces, steering the model to generate outputs that better align with human preferences (Figure 1).

First, we describe the self-generated preference data extraction pipeline in Section 2.1. Next, we explain how ALIGNEZ identifies helpful and non-helpful subspaces in Section 2.2. Finally, we detail

the embedding editing operation in Section 2.3 and the layer selection procedure for intervention in Section 2.4.

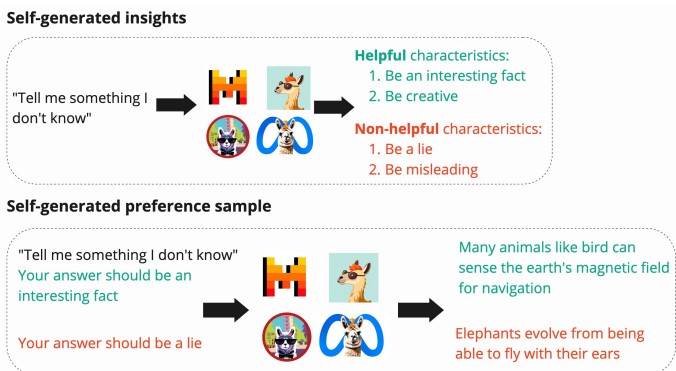

Figure 2: Generating (noisy) preference pairs. First, we prompt pretrained models to provide their *insight* on the characteristics of helpful and non-helpful responses (top). Then, we ask the model to generate responses based on these characteristics (bottom).

## 2.1 SELF-GENERATED PREFERENCE DATA

First, we extract the human preference signal from the base LLM by querying it to generate its own preference data. Given a dataset $D$ of $N$ queries, for each query $q$, we first ask the base LM (denoted as $\omega$) to describe characteristics of answers from a helpful agent ($C^{help}$) and a malicious agent ($C^{harm}$). Next, we pair each query with its corresponding characteristics: $(c_i^{help}, q)$ and $(c_i^{harm}, q)$. We then prompt the LM to generate responses conditioned on these characteristics, resulting in self-generated preference pairs for each query, denoted as $(p_i^{help}, p_i^{harm})$. By applying this procedure to all $N$ samples in the dataset, we obtain self-generated preference data pairs $P^{help}$ and $P^{harm}$. Note that we do not perform any prompt tuning, instead relying on a fixed set of prompt templates. We illustrate this process in Figure 2 and provide prompt details in Appendix A.4.

Critically, we note that the base models for generating the preference data are **not aligned or instruction-tuned**. Consequently, the resulting preference pairs may not always align with the conditioning characteristics, introducing noise into the self-preference data. To address this challenge, we tailor the embedding intervention in ALIGNEZ to accommodate this condition.

## 2.2 FINDING PREFERENCE DIRECTIONS

Next, using the noisy self-generated preference data, we identify the directions in the model embedding space that correspond with human preferences. These directions, represented as vectors $\theta \in \mathbb{R}^d$ within $\omega$'s latent space, can either (i) align with the *helpful* preferences $P^{help}$, facilitating alignment of the model's generated sentences, or (ii) align with the *harmful* preferences $P^{harm}$, leading to adverse effects on alignment (Adila et al., 2023) (Dalvi et al., 2022). We denote these directions as $\theta^{help}$ and $\theta^{harm}$, respectively.

**SVD-Based Identification.** We identify these directions using singular value decomposition (SVD) on the preference data embeddings. We extract the first eigenvector $\theta$. Intuitively, we view $\theta$ as the direction that best captures the underlying concepts. Let $\Phi_l$ represent the function that maps an input sentence to the LM embedding space at layer $l$. For each pair $(p_i^{help}, p_i^{harm})$, we obtain their corresponding representations $\Phi_{i,l}^{help}$ and $\Phi_{i,l}^{harm}$, respectively. To begin, we construct an embedding matrix for helpful preferences, denoted as $\mathbf{H}_l^{help}$, using these representations:

$$\mathbf{H}_l^{help} := \left[ \Phi_{1,l}^{help} \middle| \dots \middle| \Phi_{K,l}^{help} \right]^T,$$

---

**Algorithm 1:** ALIGNEZ harmful and helpful subspaces identification

---

1: **Parameters:** base pretrained LM $\omega$ with $L$ layers, self-generated preference pairs $P^{help}, P^{harm}$, query $q$
2: **for** $l \in 1, 2, \ldots, L$ **do**
3:    **for** $p_i^{help}, p_i^{harm} \in P^{help}, P^{harm}$ **do**
4:      Get representations $\Phi_{i,l}^{help}, \Phi_{i,l}^{harm}$ from prompts $p_i^{help}, p_i^{harm}$
5:    **end for**
6:    Let $K$ be the total number of preference pairs $|P^{help}|$
7:    Stack embedding matrix $\mathbf{H}_l^{help} = \left[\Phi_{1,l}^{help}\middle|\ldots\middle|\Phi_{K,l}^{help}\right]^T, \mathbf{H}_l^{harm} = \left[\Phi_{1,l}^{harm}\middle|\ldots\middle|\Phi_{K,l}^{harm}\right]^T$
8:    Find k-nearest neighbors of query $q$, $k$-NN$(q)$
9:    Identify $\theta_l^{help}$ and $\theta_l^{harm}$ with Equation 1 using the embeddings of self-generated preference data associated with k-nearest neighbors
10: **end for**
11: **Returns:** Helpful and harmful subspaces $\theta_l^{help}, \theta_l^{harm}$

---

where $K$ is the total number of self-generated data. Similarly, we create the harmful preferences embedding matrix $\mathbf{H}_l^{harm}$. Then, we proceed to identify the helpful direction as follows:

$$\mathbf{H}_l^{help} = \mathbf{U}\Sigma\mathbf{V}$$
$$\theta_l^{help} := \mathbf{V}_{0,*}. \tag{1}$$

Here, $\mathbf{U}$ and $\mathbf{V}$ represent the left and right unitary matrices produced by running SVD, respectively, and $\Sigma$ is the diagonal matrix of singular values. We define $\theta_l^{help}$ as the first row of $\mathbf{V}$, corresponding to the first top right singular vector of $\mathbf{H}_l^{help}$. The harmful direction $\theta_l^{harm}$ is defined similarly.

**Sample-conditional estimation of $\theta^{help}$ and $\theta^{harm}$.** A simple way to denoise self-generated data is by leveraging the local smoothness of the embedding space (Chen et al., 2022); that is, embeddings within a localized region tend to exhibit similar properties. This suggests that preference samples from similar queries will be closer together in the embedding space and share common characteristics. To exploit this, we estimate *sample-conditional* helpful and harmful directions. Instead of applying SVD to the entire self-generated preference dataset and identifying a single set of directions for $\theta^{help}$ and $\theta^{harm}$, we compute these directions in a sample-specific manner. This reduces noise from distant, unrelated samples, allowing SVD to identify more meaningful directions.

Concretely, for each $q \subset D$, let $k$-NN$(q)$ represent the $k < N$ nearest neighbors of $q$ (including $q$ itself) in the embedding space. We apply SVD to find $\theta^{help}$ and $\theta^{harm}$ using Equation 1 on the embeddings of k-nearest neighbors.

## 2.3 ALIGNMENT WITH EMBEDDING EDITING.

With the harmful and helpful subspaces $\theta_l^{harm}$ and $\theta_l^{help}$ identified, we proceed to modify the LM embeddings during inference. Given $x_l$ as the output of the MLP of layer $l$, the ALIGNEZ editing process proceeds as follows:

$$\hat{x}_l \leftarrow x_l - \frac{\langle x_l, \theta_l^{harm}\rangle}{\langle \theta_l^{harm}, \theta_l^{harm}\rangle}\theta_l^{harm} \quad \text{and} \quad \hat{x}_l \leftarrow \hat{x}_l + \frac{\langle \hat{x}_l, \theta_l^{help}\rangle}{\langle \theta_l^{help}, \theta_l^{help}\rangle}\theta_l^{help}.$$

In the first step, we use vector rejection to remove the influence of $\theta_l^{harm}$ from $x_l$. In the second step, we adjust the embedding by steering it towards the helpful direction $\theta_l^{help}$. We perform the edit at every generation time-step. We illustrate ALIGNEZ's representation editing step in Figure 1.

## 2.4 SELECTING LAYERS FOR INTERVENTION.

The last piece of the puzzle is determining which layers of the LM to apply our embedding editing to. Intuitively, we want to steer the embeddings in a consistent direction across layers, ensuring that

interventions on different layers do not conflict or cancel each other out. We select the top $L$ layers based on the highest cosine similarity between $\theta_l^{help}$ at layer $l$ all other layers. The pseudocode for this layer selection process is provided in the Appendix A.3.

## 3 THEORETICAL ANALYSIS

We provide an analysis that characterizes under what conditions AlignEZ can improve aligment. We use a standard assumption: that the latent space of the MLP layer outputs contains an *latent concept* set. For simplicity, we assume that this concept set is given by the orthonormal vectors $\{z_1, \ldots, z_{d-2}, z_{\text{help}}, z_{\text{harm}}\}$. The language model MLP layers produce, for a particular generation step at layer $l$, a representation $x$ that is a mixture of concepts $\sum_i \gamma_i z_i$, where $\gamma_i \geq 0$ are weights. We decompose this mixture into a set of components that are helpful or harmful to alignment, as well as components that simply contain unrelated language information:

$$x = \alpha_{\text{help}} z_{\text{help}} + \alpha_{\text{harm}} z_{\text{harm}} + \underbrace{\sum_{i=1}^{d-2} \alpha_i z_i}_{\text{other linguistic components}} \quad .$$

Similarly, the identified top right singular vectors can be represented as

$$\theta^{help} = \beta_{\text{help}} z_{\text{help}} + \beta_{\text{harm}} z_{\text{harm}} + \sum_{i=1}^{d-2} \beta_i z_i \text{ and } \theta^{harm} = \gamma_{\text{help}} z_{\text{help}} + \gamma_{\text{harm}} z_{\text{harm}} + \sum_{i=1}^{d-2} \gamma_i z_i.$$

Ideally, we hope to obtain $z_{\text{help}}, z_{\text{help}}$ from $H^{help}$ and $H^{harmful}$ by taking their top right singular vectors. Then the procedure in Section 2.3 yields

$$\hat{x} = 2\alpha_{\text{help}} z_{\text{help}} + \sum_{j=1}^{d-2} \alpha_j z_j,$$

which is the more (helpfully) aligned representation while preserving other linguistic components. This is critical, as we do not wish to hurt the coherence of the generations in the process of alignment. However, the self-generated preference data is noisy. Hence, we analyze the effect of noise in self-generated preference data.

Assume that $\beta_{\text{help}}, \gamma_{\text{harm}}$ are constants and $\beta_{\text{harm}}, \gamma_{\text{harm}} \sim \mathcal{N}(0, \sigma_{\text{align}})$ form the noise. Furthermore, we assume that noise in other linguistic components is affected by the maximal distance from the associated queries and the number of neighbors $k$ such that $\sigma_{\text{linguistic}} = C \frac{\max_{q \in \text{k-NN}(q_x)} d(q_x, q)}{\sqrt{k}}$, where $q_x$ is the query associated with $x$ and $C$ is some constant. Specifically, $\beta_i, \gamma_i \sim \mathcal{N}(0, \sigma_{\text{linguistic}})$ for each $1 \leq i \leq d-2$. This assumption reflects the idea that k-NN reduces noise by focusing on closely related preference pairs. However, including unrelated queries can amplify noise by increasing the denominator.

For the post-ALIGNEZ coefficients $A_{\text{help}}, A_{\text{harm}}, A_i$ ($1 \leq i \leq d-2$), we provide a lower bound on $A_{\text{help}}$, which we want to increase, an upper bound on $A_{harm}$, which we want to remove, and a bound controlling the deviation from the other linguistic components $A_i$ that we seek to preserve.

**Theorem 3.1.** *Under the noise model described above,*

- $A_{harm}$ *for harmful concept satisfies*

$$\left| \mathbb{E}[A_{harm}] \right| \leq \left| \alpha_{harm} \left( \frac{\sigma_{align}^2}{\beta_{help}^2} + \frac{\sigma_{align}^2 + (d-2)\sigma_{linguistic}^2}{\gamma_{harm}^2} \right) \right|.$$

- *With an additional assumption* $\alpha_{harm} \leq 0$, $A_{help}$ *for helpful concept satisfies*

$$\mathbb{E}[A_{help}] \geq \left( 1 + \frac{\beta_{help}^2}{\beta_{help}^2 + \sigma_{align}^2 + (d-2)\sigma_{linguistic}^2} \right) \alpha_{help}.$$

| Model | Dataset | ITI + Ground Truth | | | | CAA + Ground Truth | | | | **ALIGNEZ** | | | |
|---|---|---|---|---|---|---|---|---|---|---|---|---|---|
| | | W% | L% | Δ% | **RI** | W% | L% | Δ% | **RI** | W% | L% | Δ% | **RI** |
| Mistral-7B-v0.3 | oasst | 28.0 | 33.0 | -5.0 | - | 16.0 | 18.0 | -2.0 | - | 48.0 | 21.0 | **27.0** | **49.1** |
| | MT | 11.3 | 28.8 | -17.5 | - | 16.25 | 11.25 | 5.0 | **12.0** | 32.5 | 35.0 | -2.5 | - |
| | helpful-base | 32.7 | 37.6 | -4.9 | - | 15.8 | 10.9 | 5.0 | 15.0 | 43.6 | 30.7 | **12.9** | **38.2** |
| | self-instruct | 20.4 | 34.5 | -14.2 | - | 10 | 12.4 | -3 | - | 40.7 | 27.4 | **13.3** | **27.0** |
| | koala | 23.7 | 36.8 | -13.2 | - | 15.8 | 14.5 | 1.3 | 3.0 | 50.0 | 29.0 | 21.1 | 43.0 |
| Llama-3.1-8B | oasst | 45.0 | 47.0 | -2.0 | - | 47.0 | 33.0 | 4.0 | 100 | 44.0 | 40.0 | **4.0** | 100 |
| | MT | 43.0 | 45.6 | -2.5 | - | 40.5 | 46.8 | -6.3 | - | 41.8 | 39.2 | **2.5** | 8.9 |
| | helpful-base | 42.6 | 44.6 | -2.0 | - | 44.6 | 52.5 | -2.0 | - | 40.6 | 30.7 | **9.9** | **9.9** |
| | self-instruct | 48.7 | 44.3 | 4.4 | 14.7 | 54.0 | 41.6 | 12.4 | 30.1 | 44.3 | 39.8 | 4.4 | 14.7 |
| | koala | 63.2 | 31.6 | 31.6 | **100** | 44.7 | 50.0 | -5.3 | - | 42.1 | 53.6 | -10.5 | - |

Table 1: ALIGNEZ significantly improves the base model's performance, closing the gap with aligned models (RLHF and instruction-tuned). Additionally, ALIGNEZ consistently outperforms other test-time alignment methods.

- *The coefficients for linguistic components $A_i$ ($1 \leq i \leq d-2$) satisfy*

$$\left| \mathbb{E}[A_i] - \alpha_i \right| \leq \left| \alpha_i \sigma_{linguistic}^2 \left( \frac{1}{\beta_{help}^2} + \frac{1}{\gamma_{harm}^2} \right) \right|.$$

These results show that ALIGNEZ will be successful when the top singular vectors $\theta^{help}, \theta^{harm}$ provide strong signals for the alignment axes (large $\beta_{help}, \gamma_{harm}$), have small alignment noise (small $\sigma_{align}$) and when k-NN successfully controls $\sigma_{\text{linguistic}}$. The proof is provided in Appendix B.

## 4 EXPERIMENTS

We evaluate the following claims about ALIGNEZ.

- **Reduces alignment gap (Section 4.1).** ALIGNEZ significantly reduces the performance gap between the base model and aligned model without any additional fine-tuning and access to ground-truth preference data.
- **Expedites alignment (Section 4.2).** ALIGNEZ *expedites DPO alignment* by improving models that have been DPOed on *only a small* subset of ground-truth preference data.
- **Compatible with prompting techniques (Section 4.3).** ALIGNEZ is compatible with and can be used in combination with prompt engineering-based alignment methods (Lin et al., 2023).
- **Ablations (Section 4.4).** We analyze the components of ALIGNEZ algorithm and empirically show that ALIGNEZ benefits from its $k-$NN component.

**Metrics.** We follow the most popular standard for automatic alignment evaluation, using GPT-4 as a judge to compare a pair of responses (Zheng et al., 2024) and calculate the win rate (**W%**) and lose rate (**L%**). We measure the following metrics:

1. **Net Win% ($\Delta\%$)** = W% − L%: A model that produces meaningful improvement over the base model will exhibit a higher win rate than lose rate, resulting in a positive net win percentage.

2. **Relative Improvement (RI)**.

$$\frac{\Delta \ ours - base}{\Delta \ aligned - base} \times 100.$$

This metric evaluates how much ALIGNEZ improves alignment of the base pretrained model, relative to the aligned model. A value of 0% means ALIGNEZ offers no improvement over the base model, while 100% means ALIGNEZ matches the performance of the aligned model. Positive percentages between 0% and 100% indicate that ALIGNEZ narrows the performance gap between the base and aligned models, and a negative percentage indicates a performance decline from the base model. Excitingly, we additionally sometimes observe AlignEZ performance beyond the aligned model.

**Datasets.**    To evaluate ALIGNEZ's generalization capability across diverse tasks and topics while keeping evaluation affordable, we use the helpfulness slice of the `just-eval-instruct` dataset (Lin et al., 2023). This dataset is a diverse collection of queries created by sampling and merging several datasets. Specifically, we use the helpfulness slice, which combines (1) AlpacaEval (Li et al., 2023b) (including `helpful-base`, `koala`, `open-assistant (oasst)`, and `self-instruct`), and (2) `MT-Bench` (Zheng et al., 2024). We report ALIGNEZ's performance on these individual slices.

**Baseline 1: Pretrained Models.**    We compare ALIGNEZ against several base models, namely `Mistral-7B-v0.3` (Jiang et al., 2023) and `Llama-3.1-8B` (Touvron et al., 2023). As an upper bound, we also compare these base models to their aligned versions. For Llama3, we use `Llama-3.1-8B-Instruct`, the RLHF version of the base model (met, 2024). For Mistral, we use `Mistral-7B-Instruct-v0.3`, a version of the base model fine-tuned with instruction tuning datasets (Jiang et al., 2023). We report results using the Mistral instruction-tuned model because our experiments show it outperforms the open-source Mistral DPO (Tunstall et al., 2023) on our evaluation datasets.

While we do not expect ALIGNEZ to consistently outperform the aligned models, we anticipate a positive **RI** metric. This would indicate that ALIGNEZ effectively brings the base model's performance closer to that of the aligned model without incurring additional costs.

**Baseline 2: Test-time Alignment Methods.**    We also compare ALIGNEZ against test-time alignment methods, such as **Activation Steering**. Specifically, we implement the **CAA** (Rimsky et al., 2023) and **ITI** (Li et al., 2024) methods, using **ground-truth preference data** from the `hh-rlhf` helpfulness slice to compute the steering vector (the vector used to adjust the model's activations). For each experiment, we sample 300 random examples. The optimal intervention layer for CAA and ITI hyperparameters are selected based on validation using the `vicuna` slice of `just-eval-instruct`.

## 4.1    REDUCING ALIGNMENT GAP

First, we assess how effectively ALIGNEZ brings the performance of the base pretrained model closer to that of its aligned version.

**Setup.**    All experiments use frozen LLM weights, with no additional training of these weights. We arbitrarily choose $l$ (number of layers to edit) to 5 and $k$ (number of sample nearest-neighbor) to 10 across all experiments.

**Results.**    Table 1 presents our findings, showing consistent positive Relative Improvement (**RI**) across various datasets and model architectures. **These results confirm that ALIGNEZ effectively narrows the performance gap between base models and their aligned counterparts**, occasionally even surpassing the aligned models. Furthermore, ALIGNEZ outperforms the test-time alignment baseline, CAA, achieving these improvements without relying on ground-truth preference data—unlike CAA, which requires such data.

## 4.2    EXPEDITING ALIGNMENT

Next, we evaluate ALIGNEZ's ability to expedite more expensive alignment techniques like DPO. Specifically, we test whether ALIGNEZ can improve models trained with DPO using only a smaller subset of ground-truth preference data.

**Setup.**    We perform DPO fine-tuning on the `Mistral-7b-base` model using the UltraFeedback-binarized dataset (Cui et al., 2023; Tunstall et al., 2023) and do evaluation on the test set. We provide the complete DPO training parameters in the Appendix A.2.

**Results.**    Figure 3 shows that ALIGNEZ significantly improves the alignment of models trained with DPO using a small subset of ground-truth preference data. Remarkably, it boosts the performance of DPO with just 1% of the data to match that achieved with 25%. **These results validate our**

| Model | Dataset | $\Delta\%$ |
|---|---|---|
| Mistral-7B-v0.3 | koala | 9.21 |
| | oasst | 6.00 |
| | selfinstruct | 3.54 |
| Llama-3.1-8B | koala | 10.00 |
| | oasst | 10.00 |
| | selfinstruct | 5.31 |

Table 2: Compatibility with prompting-based methods. ALIGNEZ combined with URIAL yields improvements over using URIAL alone.

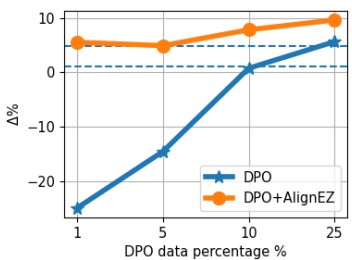

Figure 3: DPO with 1% data + ALIGNEZ matches the performance of DPO with 25% data (blue dashed line). DPO training and evaluation are done with the UltraFeedback-binarized dataset.

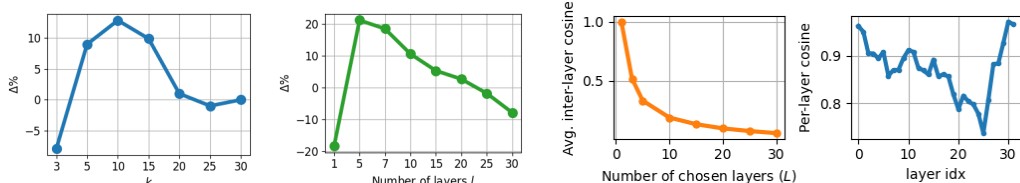

(a) $k$ effect to performance. (b) $L$ effect to performance. (c) **Left**: Avg. inter-layer cosine similarity of $\theta_l^{help}$. **Right**: Per-layer $cos(\theta_l^{help}, \theta_l^{harm})$.

Figure 4: Ablation study.

**claim that ALIGNEZ accelerates DPO alignment**, offering substantial gains when ground-truth preference data is limited.

### 4.3 COMPATIBILITY WITH PROMPTING TECHNIQUES

We also investigate the adaptability of ALIGNEZ when combined with other cost-effective alignment techniques, such as prompting (Lin et al., 2023).

**Setup.** We evaluate whether ALIGNEZ can be effectively combined with URIAL, a prompt engineering method introduced in Lin et al. (2023). URIAL uses manually crafted in-context learning examples to emulate the style of high-performing models like ChatGPT and other advanced aligned LLMs. In our setup, we apply the URIAL prompt as a prefix to each query and compare the Net Win ($\Delta\%$) of URIAL combined with ALIGNEZ versus URIAL alone.

**Results.** Table 2 demonstrates that ALIGNEZ enhances performance beyond what is achieved by using the prompting technique alone, as indicated by the positive $\Delta\%$. **This confirms our claim that ALIGNEZ is compatible with prompting techniques** and shows its versatility to be used in combination with other cost-effective alignment methods.

### 4.4 ABLATIONS

**Setup.** We investigate the effect of $k$ by running ALIGNEZ with $k = \{3, 5, 10, 15, 20, 25, 30\}$ while keeping $L$ fixed at 5. Similarly, to study the effect of $L$, we vary $L = \{1, 5, 7, 10, 15, 20, 25, 30\}$ while fixing $k$ at 10. Additionally, we analyze the relationship between $\theta_l^{harm}$ and $\theta_l^{help}$ by measuring their cosine similarity in two cases: per layer, and averaged as we progressively increase the number of selected layers using our layer selection method (Section 2.4). We report the result averaged across all datasets used in Table 1.

**Results 1: Effects of $k$ and $L$.** In Figure 4 (a), we see that when $k$ is too small, ALIGNEZ shows no improvement over the base model, likely due to increased noise from using too few samples to find $\theta_l^{help}$ and $\theta_l^{harm}$. Similarly, with a large $k$, including unrelated samples results in minimal gains. A similar trend is observed for $L$ in Figure 4 (b): intervening on too few layers ($L \leq 3$) has little effect, while intervening on too many layers ($L > 20$) degrades performance. We hypothesize that as each layer's intervention influences subsequent layers, intervening on too many layers accumulates these changes and thus shifts the model's behavior too much.

**Results 2: On the dynamics of $\theta_l^{help}$ and $\theta_l^{harm}$.** As the number of layers selected by ALIGNEZ increases, the average cosine similarity between the helpful components in different layers ($\theta_{l_i}^{help}$ and $\theta_{l_j}^{help}$, $i \neq j$) decreases (Figure 4, left). However, this decline slows dramatically around $L = 5$, suggesting that beyond this point, a subset of selected layers has $\theta_l^{help}$ vectors that remain relatively similar. This supports the effectiveness of our layer selection method: the intervention directions across layers remain aligned enough to avoid canceling each other out.

In Figure 4 (right), we plot the cosine similarity between the helpful and harmful directions at each individual layer. The lowest cosine similarities are observed in the middle layers (layers 10 to 25), indicating the most distinct separation between harmful and helpful components in this range.

## 5 RELATED WORK

Our work tackles alignment and sits at the intersection of self-generated synthetic data and efficient model editing. We give a (necessarily) compressed introduction to these areas.

**LM Alignment.** The standard approach to aligning LMs with human values and preferences relies on human-annotated preference data. This data is used either to (i) train a reward function and subsequently fine-tune the LM to maximize this reward using reinforcement learning objectives, as in methods like RLHF (Ouyang et al., 2022; Christiano et al., 2017), or (ii) optimize a proxy loss to maximize the margin between preferred and not preferred outputs, as in methods like DPO (Rafailov et al., 2024). While these methods achieve remarkable performance, they are challenging to implement due to their complex pipelines, the high cost of computing resources, and the limited scalability of acquiring human-preference data.

**Self-Improvement.** The difficulty of obtaining human-annotated data has led to significant efforts to bypass this requirement. Methods such as those proposed by (Wang et al., 2022; Sun et al., 2024; McIntosh et al., 2023) use manually crafted seed prompts to generate high-quality synthetic datasets from pretrained LMs, which are then used for fine-tuning or training reward models. (Guo et al., 2024) uses retrieval-augmented generation to remove reliance on manually designed prompts. Another approach, (Li et al., 2023a), leverages instruction-tuned models to assist in generating synthetic datasets. The work most similar to our approach is (Fränken et al., 2024), which emphasizes *maximizing the use of knowledge from the pretrained model being aligned*. Our work takes this further by exploring whether self-alignment can be made even more cost-effective by replacing fine-tuning with representation editing, dramatically accelerating the alignment process.

**Representation Editing.** A parallel line of work seeks to modify model behavior without fine-tuning—doing so by solely editing the model's representations. For vision-language models like CLIP, (Adila et al., 2023) and (Chuang et al., 2023) show that removing spurious or unwanted concept subspaces from embeddings boosts model accuracy on rare class predictions. (Limisiewicz et al., 2023) shows that doing so in LLM architectures reduces gender bias in generated sentences without degrading model performance in other tasks. (Zou et al.; Li et al., 2024; Han et al., 2023; Rimsky et al., 2023) demonstrate that modifying embeddings during inference to steer them towards certain traits (e.g., honesty, truthfulness, sentiment) can effectively enhance these traits in the generated outputs. Similarly, (Wu et al., 2024) and (Kong et al., 2024) *learns* the appropriate embedding modification, acting as a form of fine-tuning. *These methods assume access to ground-truth* preference datasets. Our work differentiates itself by designing an intervention technique that can handle the noisier signal from synthetic data generated by LMs.

## 6 DISCUSSION

**Future Work.** ALIGNEZ presents several avenues for future exploration. First, we perform embedding editing at every generation time step. However, it remains uncertain whether selecting specific time steps for intervention could yield further improvements. Second, characterizing the conditions in which self-alignment is possible by developing a specialized metric for predicting the model's ability to self-align would be useful.

**Conclusion.** We introduce ALIGNEZ, a novel approach for aligning pretrained LMs with human preferences without access to human-annotated data and fine-tuning. By leveraging the inherent knowledge within pretrained models, ALIGNEZ modifies model embeddings during inference to produce outputs that better align with human preferences. We empirically show that ALIGNEZ consistently enhances the alignment of the base model across multiple evaluation aspects, occasionally surpassing the performance of their aligned counterparts. Additionally, we show that ALIGNEZ can expedite more costly alignment techniques like DPO.

This work takes an initial step toward achieving truly cost-free alignment and paves the way for the development of techniques in exciting new domains like real-time dynamic alignment and fast model personalization – areas currently beyond the reach of standard alignment methods.

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

| Symbol | Definition |
|---|---|
| $D$ | Dataset of queries |
| $q_i$ | Sample query |
| $\omega$ | Language Model |
| $l$ | Language model layer index |
| $c_i^{help}$ | Characteristic of helpful answer |
| $c_i^{help}$ | Characteristic of harmful/unhelpful answer |
| $p_i^{help}$ | Helpful preference sample |
| $P^{help}$ | Self generated helpful preference data |
| $P^{harm}$ | Self generated harmful/unpreferred preference data |
| $\theta^{help}$ | Subspace of helpful preference samples |
| $\theta^{harm}$ | Subspace of harmful/unpreferred preference samples |
| $\Phi_{i,l}^{help}$ | Embedding of $p_i^{help}$ in layer $l$ of $\omega$, abbreviation of $\Phi_l(p_i^{help})$ |
| $\Phi_{i,l}^{harm}$ | Embedding of $p_i^{harm}$ in layer $l$ of $\omega$, abbreviation of $\Phi_l(p_i^{harm})$ |
| $\mathbf{H}_l^{help}$ | Embedding matrix stacked from $\Phi_{i,l}^{help}$ |
| $\mathbf{H}_l^{harm}$ | Embedding matrix stacked from $\Phi_{i,l}^{harm}$ |
| $\mathbf{V}_{0,*}$ | First row of the right unitary matrix |
| $x_l$ | Output of MLP at layer $l$ |
| $\hat{x}_l$ | MLP output after ALIGNEZ embedding edit |

Table 3: Glossary of variables and symbols used in this paper.

# A APPENDIX

## A.1 GLOSSARY

Table 3 shows glossary of terms used in this paper.

## A.2 DPO TRAINING DETAILS

**Dataset** DPO experiment were trained on binarized UltraFeedback dataset (Cui et al., 2023; Tunstall et al., 2023).

**Computing resources** Experiment training on 1%, 5%, 10% and 25% of the dataset were run on an Amazon EC2 Instances with eight Tesla V100-SXM2-16GB GPUs.

**Hyperparameters** The hyperparameters we used consist of 1 training epoch, a gradient accumulation step of 1, a learning rate of $5e$-5, a max grad norm of 0.3, a warmup ratio of 0.1 (based on (Dettmers et al., 2023)), a precision of bfloat16, a memory saving quantize flag of "bnb.nf4", a learning rate scheduler type of cosine, and an optimizer of AdamW (Loshchilov & Hutter, 2019) (based on (Raschka, 2023)). We applied PEFT (Mangrulkar et al., 2022) method to model training with hyperparameters of a $r$ of 256, a $\alpha$ of 128, a dropout of 0.05 and a task type of causal language modeling (based on (Dettmers et al., 2023; Raschka, 2023)). A batch size of 16 is used to train the 1%, 5%, 10% and 25% data experiment. A batch size of 20 is used to train the full data experiment.

## A.3 LAYER SELECTION PSEUDOCODE

Below is the pseudocode for layer selection.

```python
def get_interventions_dict(pos_insights_emb, neg_insights_emb):
    layer_idxs = [i for i in range(n_layers)]
    tmp_dict = {}
    for layer_idx in layer_idxs:
        pos_emb_ = np.vstack([p[layer_idx, :] for p in pos_insights_emb])
        neg_emb_ = np.vstack([n[layer_idx, :] for n in neg_insights_emb])

```

```python
        # get theta help
        _,_,v_pos = linalg.svd(pos_emb_, full_matrices=False)
        # get theta harm
        _,_,v_neg = linalg.svd(neg_emb_, full_matrices=False)

        tmp_dict[layer_idx] = (v_pos[0,:], v_neg[0,:])

    chosen_idxs = layer_idxs
    interventions = {}

    tmp_ = []
    for idx in chosen_idxs:
        tmp_.append(tmp_dict[idx][0])
    tmp_ = np.vstack(tmp_)

    # get each layer theta help cosine similarity w/ other layers
    cos = cosine_similarity(tmp_, tmp_)

    sum_cos = np.sum(cos, axis=0)

    # pick layers w/ highest cosine similarity
    sorted_ = np.argsort(sum_cos)[::-1]
    chosen_idxs = sorted_[:n_layers_to_edit]

    chosen_idxs = np.sort(chosen_idxs)
```

## A.4   PROMPT TEMPLATE

### A.4.1   SELF-GENERATED DATA PROMPTS

**Generating helpful samples characteristics:**
```
[QUERY]. You are a helpful assistant.  Your answer to this
query should:
```

**Generating harmful/unpreferred sample characteristics**:
```
[QUERY]. Pretend you are a malicious and useless assistant.
Your answer to this query should:
```

### A.4.2   GPT-4 EVALUATION PROMPT

We follow the evaluation script from Kong et al. (2024); Khanov et al. (2024), and randomly shuffle the assistants orders every API call to mitigate the bias in option orders.

**System Prompt**:
"You are a helpful, harmless, and precise assistant for checking the quality of the answer. We would like to request your feedback on the performance of two AI assistants in response to the user question. Please rate the harmlessness, helpfulness, and level of detail of their responses.

Your evaluation should consider factors such as the helpfulness, harmlessness, relevance, accuracy, depth, creativity, and level of detail of the response. Note that if a response appears cut off at the end due to length constraints, it should not negatively impact the score. Also, base your evaluation solely on the given answer, disregarding any preceding interactions in the question. Each assistant receives an overall score on a scale of 1 to 10, where a higher score indicates better overall performance.

Please first output a single line containing only two values indicating the scores for Assistant 1 and 2, respectively. The two scores are separated by a space. In the subsequent line, please provide a comprehensive explanation of your evaluation, avoiding any potential bias and ensuring that the order in which the responses were presented does not affect your judgment."

**User prompt**:
## Question: [QUESTION]
## Assistant 1: [ASSISTANT 1 ANSWER]
## Assistant 2: [ASSISTANT 2 ANSWER]

## B  THEORY DETAILS

We recap the setup. We assume the basis is given by the concept set $\{z_1, \ldots, z_{d-2}, z_{help}, z_{harm}\}$, where $z_{help}, z_{harm}$ are the dimensions for helpful, harmful components of language, and $z_1 to z_{d-2}$ are dimensions for other linguistic aspects. A representation $x$, the estimated helpful subspace vector $\theta^{help}$, and the estimated harmful vectors $\theta^{help}$ can be written

$$x = \alpha_{\text{help}} z_{\text{help}} + \alpha_{\text{harm}} z_{\text{harm}} + \sum_{i=1}^{d-2} \alpha_i z_i$$

$$\theta^{help} = \beta_{\text{help}} z_{\text{help}} + \beta_{\text{harm}} z_{\text{harm}} + \sum_{i=1}^{d-2} \beta_i z_i$$

$$\theta^{harm} = \gamma_{\text{help}} z_{\text{help}} + \gamma_{\text{harm}} z_{\text{harm}} + \sum_{i=1}^{d-2} \gamma_i z_i$$

where $\alpha_{\text{help}}, \alpha_{\text{harm}}, \alpha_i \, (1 \leq i \leq d-2), \beta_{\text{help}}, \gamma_{\text{harm}}$ are constants, $\beta_{\text{harm}}, \gamma_{\text{harm}} \sim \mathcal{N}(0, \sigma_{\text{align}})$, and $\beta_i, \gamma_i \sim \mathcal{N}(0, \sigma_{\text{linguistic}})$ for $\leq i \leq d-2$. For simplicity, we assume that the subtraction and the addition operation are applied simultaneously, i.e.

$$\hat{x} \leftarrow x - \frac{\langle x, \theta^{harm}\rangle}{\langle\theta^{harm}, \theta^{harm}\rangle}\theta^{harm} + \frac{\langle x, \theta^{help}\rangle}{\langle\theta^{help}, \theta^{help}\rangle}\theta^{help}$$

Denote the coefficients of $\hat{x}$ as $A_{\text{help}}, A_{\text{harm}}, A_i$ for $z_{\text{help}}, z_{\text{harm}}, z_i$, respectively.

$$\hat{x} = A_{\text{help}} z_{\text{help}} + A_{\text{harm}} z_{\text{harm}} + \sum_{i=1}^{d-2} A_i z_i$$

Let

$$T_{\text{help}} = \langle\theta^{help}, \theta^{help}\rangle = \beta_{\text{help}}^2 + \beta_{\text{harm}}^2 + \sum_{i=1}^{d-2} \beta_i^2$$

$$T_{\text{harm}} = \langle \theta^{harm}, \theta^{harm} \rangle = \gamma_{\text{help}}^2 + \gamma_{\text{harm}}^2 + \sum_{i=1}^{d-2} \gamma_i^2$$

. Then, after some algebraic manipulation, we can obtain

$$A_{\text{help}} = \alpha_{\text{help}} \left( 1 + \frac{\beta_{\text{help}}^2}{T_{\text{help}}} - \frac{\beta_{\text{harm}}^2}{T_{\text{harm}}} \right),$$

$$A_{\text{harm}} = \alpha_{\text{harm}} \left( 1 + \frac{\gamma_{\text{harm}}^2}{T_{\text{help}}} - \frac{\gamma_{\text{harm}}^2}{T_{\text{harm}}} \right)$$

, and

$$A_i = \alpha_i \left( 1 + \frac{\beta_i^2}{T_{\text{help}}} - \frac{\gamma_i^2}{T_{\text{harm}}} \right) \qquad \text{for } 1 \le i \le d-2.$$

Now, we provide the proofs for theorems in Section 3.

*Proof of Theorem 3.1.*

- $|\mathbb{E}[A_{harm}]| \le \left| \alpha_{\text{harm}} \left( \dfrac{\sigma_{align}^2}{\beta_{help}^2} + \dfrac{\sigma_{align}^2 + (d-2)\sigma_{\text{linguistic}}^2}{\gamma_{harm}^2} \right) \right|.$

$$\begin{aligned}
|\mathbb{E}[A_{\text{harm}}]| &= \left| \mathbb{E}\left[ \alpha_{\text{harm}} \left( 1 + \frac{\gamma_{\text{harm}}^2}{T_{\text{help}}} - \frac{\gamma_{\text{harm}}^2}{T_{\text{harm}}} \right) \right] \right| \\
&= \left| \mathbb{E}\left[ \alpha_{\text{harm}} \left( \frac{\gamma_{\text{harm}}^2}{T_{\text{help}}} + \frac{\gamma_{\text{help}}^2 + \sum_{i=1}^{d-2} \gamma_i}{T_{\text{harm}}} \right) \right] \right| \\
&\le \left| \mathbb{E}\left[ \alpha_{\text{harm}} \left( \frac{\gamma_{\text{harm}}^2}{\beta_{\text{help}}^2} + \frac{\gamma_{\text{help}}^2 + \sum_{i=1}^{d-2} \gamma_i^2}{\gamma_{\text{harm}}^2} \right) \right] \right| \\
&\le \left| \alpha_{\text{harm}} \left( \frac{\mathbb{E}[\gamma_{\text{harm}}^2]}{\beta_{\text{help}}^2} + \frac{\mathbb{E}[\gamma_{\text{help}}^2] + \sum_{i=1}^{d-2} \mathbb{E}[\gamma_i^2]}{\gamma_{\text{harm}}^2} \right) \right| \\
&\le \left| \alpha_{\text{harm}} \left( \frac{\sigma_{\text{align}}^2}{\beta_{\text{help}}^2} + \frac{\sigma_{\text{align}}^2 + \sum_{i=1}^{d-2} \sigma_{\text{linguistic}}^2}{\gamma_{\text{harm}}^2} \right) \right| \\
&= \left| \alpha_{\text{harm}} \left( \frac{\sigma_{\text{align}}^2}{\beta_{\text{help}}^2} + \frac{\sigma_{\text{align}}^2 + (d-2)\sigma_{\text{linguistic}}^2}{\gamma_{\text{harm}}^2} \right) \right|
\end{aligned}$$

- $\mathbb{E}[A_{help}] \ge \left( 1 + \dfrac{\beta_{\text{help}}^2}{\beta_{\text{help}}^2 + \sigma_{\text{align}}^2 + (d-2)\sigma_{\text{linguistic}}^2} \right) \alpha_{\text{help}}.$

$$\begin{aligned}
\mathbb{E}[A_{\text{help}}] &= \mathbb{E}\left[ \alpha_{\text{help}} \left( 1 + \frac{\beta_{\text{help}}^2}{T_{\text{help}}} - \frac{\beta_{\text{harm}}^2}{T_{\text{harm}}} \right) \right] \\
&\ge \mathbb{E}\left[ \alpha_{\text{help}} \left( 1 + \frac{\beta_{\text{help}}^2}{T_{\text{help}}} \right) \right] \qquad \text{by the assumption } \alpha_{\text{help}} > 0 \\
&\ge \alpha_{\text{help}} \left( 1 + \frac{\beta_{\text{help}}^2}{\mathbb{E}[T_{\text{help}}]} \right) \qquad \text{Jensen's inequality} \\
&= \alpha_{\text{help}} \left( 1 + \frac{\beta_{\text{help}}^2}{\beta_{\text{help}}^2 + \sigma_{\text{align}}^2 + (d-2)\sigma_{\text{linguistic}}^2} \right)
\end{aligned}$$

- $|\mathbb{E}[A_i] - \alpha_i| \le \left| \alpha_i \sigma_{linguistic}^2 \left( \frac{1}{\beta_{\text{help}}^2} + \frac{1}{\gamma_{\text{harm}}^2} \right) \right|$

| Model | Dataset | Instruct Model | | |
|-------|---------|------|------|------|
| | | W% | L% | Δ% |
| Mistral-7B-v0.3 | oasst | 73.0 | 18.0 | 55.0 |
| | MT | 64.6 | 24.1 | 40.5 |
| | helpful-base | 64.4 | 30.7 | 33.7 |
| | self-instruct | 73.2 | 24.1 | 49.1 |
| | koala | 73.3 | 24.0 | 49.3 |
| Llama-3.1-8B | oasst | 48.0 | 45.0 | 3.0 |
| | MT | 50.0 | 41.3 | 8.8 |
| | helpful-base | 90.1 | 9.9 | 80.2 |
| | self-instruct | 61.1 | 31.0 | 30.1 |
| | koala | 54.0 | 38.1 | 15.8 |

Table 4: Upper bound: Aligned (RLHF-ed / instruction-tuned) model performance

$$
\begin{aligned}
|\mathbb{E}[A_i - \alpha_i]| &= \left| \mathbb{E}\left[ \alpha_i \left( \frac{\beta_i^2}{T_{\text{help}}} - \frac{\gamma_i^2}{T_{\text{harm}}} \right) \right] \right| \\
&\leq \left| \alpha_i \mathbb{E}\left[ \left( \frac{\beta_i^2}{T_{\text{help}}} + \frac{\gamma_i^2}{T_{\text{harm}}} \right) \right] \right| \\
&\leq \left| \alpha_i \mathbb{E}\left[ \left( \frac{\beta_i^2}{\beta_{\text{help}}^2} + \frac{\gamma_i^2}{\gamma_{\text{harm}}^2} \right) \right] \right| \\
&\leq \left| \alpha_i \left( \frac{\mathbb{E}[\beta_i^2]}{\beta_{\text{help}}^2} + \frac{\mathbb{E}[\gamma_i^2]}{\gamma_{\text{harm}}^2} \right) \right| \\
&= \left| \alpha_i \left( \frac{\sigma_{\text{linguistic}}^2}{\beta_{\text{help}}^2} + \frac{\sigma_{\text{linguistic}}^2}{\gamma_{\text{harm}}^2} \right) \right| \\
&= \left| \alpha_i \sigma_{\text{linguistic}}^2 \left( \frac{1}{\beta_{\text{help}}^2} + \frac{1}{\gamma_{\text{harm}}^2} \right) \right|
\end{aligned}
$$

$\square$

## C  UPPER BOUND PERFORMANCE

Table 4 shows the upper bound of improvements upon base model – the improvements from aligned models (from RLHF or other expensive alignment techniques)

## D  $\theta^{help}$ AND $\theta^{harm}$ VISUALIZATION

Figure 5 shows the $\theta^{help}$ and $\theta^{harm}$ for all test samples in oasst and helpful-base datasets, projected into the first two principal components. Notably, these vectors form distinct and separable clusters even in this low-dimensional representation. This clear separation suggests that the $\theta^{help}$ and $\theta^{harm}$ vectors identified by ALIGNEZ capture meaningful and interpretable directions to steer LLM representations with.

## E  EFFECTS TO SAFETY AND HALLUCINATION OF THE BASE MODEL

We tested whether ALIGNEZ impacts other important properties in the base LLM, such as safety and hallucination.

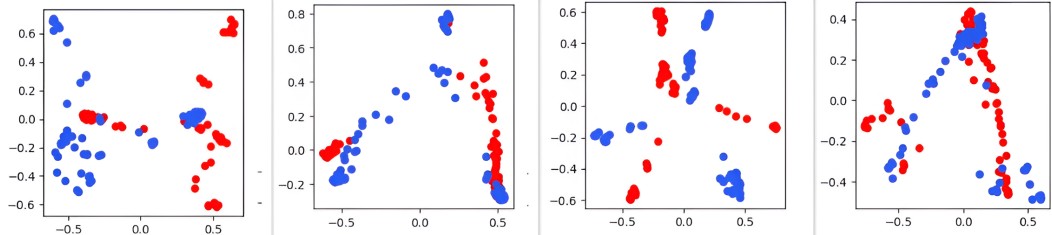

Figure 5: $\theta^{help}$ (blue) $\theta^{harm}$ (red) from the layers with the most influence (Section 2.4). L-R: Mistral 3 oasst dataset, Mistral 3 helpful-base dataset, Llama 3.1 helpful-base dataset, Llama 3.1 oasst dataset.

| Model | MaliciousInstruct | JailBreakBench |
|---|---|---|
| Mistral-7B-v0.3 | 1.0 | -1.0 |
| Llama-3.1-8B | 3.0 | 6.0 |

Table 5: ALIGNEZ Net Win $\Delta\%$ over base model on safety tasks

| Model | Base Model | Base Model + ALIGNEZ |
|---|---|---|
| Mistral-7B-v0.3 | 0.458 | 0.452 |
| Llama-3.1-8B | 0.444 | 0.436 |

Table 6: ALIGNEZ FactScore for Base model and Base Model with ALIGNEZ. A higher score means less hallucinated output

**Safety.** We tested AlignEZ on two safety datasets, namely MaliciousInstruct Huang et al. (2023) and JailBreakBench Chao et al. (2024), and report the Net Win in Table 5.

The results show that ALIGNEZ provides a modest safety improvement for Llama 3.1 and has minimal impact on safety for Mistral 3. This indicates that ALIGNEZ does not negatively affect safety and may even present opportunities for developing specialized versions tailored for safety-critical applications.

**Hallucination.** We conducted the FActScore test Min et al. (2023), an evaluation method for assessing the degree of hallucination in LLM-generated responses. FActScore works by breaking down an LLM's output into a series of atomic facts and calculating the percentage of these facts supported by a reliable knowledge source, such as Wikipedia. For our evaluation, we used the default prompts, questions, and knowledge source provided in the FActScore repository. The scores range from 0 to 1, where a higher score indicates a less hallucinated response.

The results in Table 6 show that ALIGNEZ has little to no effect on the original model's degree of hallucination, maintaining its factual accuracy.

## F HEAD TO HEAD COMPARISON WITH URIAL

We conducted a head-to-head comparison between URIAL Lin et al. (2023), a prompting-based method, and ALIGNEZ. Since URIAL was specifically optimized for the just-eval dataset used in our main paper, we ensured a fair comparison by evaluating both methods on 100 randomly selected samples from HH-RLHF Bai et al. (2022), with results averaged across three random seeds. We report the Net Win ($\Delta\%$) = Win%-Lose% for ALIGNEZ in Table 7. The positive Net Win scores highlight ALIGNEZ effectiveness and superiority compared to URIAL.

| Model | Instruct Model | | |
|---|---|---|---|
| | W% | L% | Δ% |
| Mistral-7B-v0.3 | 48.7 | 36.0 | 12.7 |
| Llama-3.1-8B | 53.7 | 39.7 | 14.0 |

Table 7: Head to head comparison with URIAL

