# OpenReview forum: "Is Free Self-Alignment Possible?"
_ICLR.cc/2025/Conference — Submitted to ICLR 2025_

### Official Review · Reviewer_yARA · 2024-10-30

**Soundness:** 2
**Presentation:** 2
**Contribution:** 2
**Rating:** 5
**Confidence:** 4

**Summary:**

This paper introduces ALIGNEZ, a method that utilizes the inherent knowledge of pretrained language models to achieve low-cost alignment using only self-generated preference data. It employs a KNN approach to identify the top n-closest preference data points for a given query, followed by SVD to extract the most relevant representations related to helpful and harmful preferences. By adjusting these embeddings during inference, ALIGNEZ effectively narrows the performance gap between pretrained and aligned models, guiding them to generate more helpful responses without requiring additional training or human-annotated data.

**Strengths:**

1. Self-Generated Data: ALIGNEZ relies solely on self-generated data from the language model, eliminating the need for additional manual annotation, which is promising for scalability.

2. Performance Improvement: Experimental results show that this alignment approach effectively reduces the performance gap compared to traditional training-based methods, without requiring any additional training.

3. Data Efficiency: By using only 1% of preference data for DPO training, ALIGNEZ achieves results comparable to those obtained with 25% of preference data, demonstrating compatibility with existing preference alignment methods.

**Weaknesses:**

1. Unclear Methodology: The description of the method's details is vague, particularly regarding the origin of queries used for generating self-generated preference data. Additionally, it's unclear which dataset was used for the experiments shown in Figures 3 and 4, and whether the reported $\Delta%$ represents an average score across multiple datasets or results from a single dataset.

2. Dependence on base LM: The base LM may generate incorrect preference data, and the paper does not clarify how it ensures that self-generated preference data accurately reflects the correct preference relationships. This lack of clarity could significantly impact the results. Furthermore, while the method shows some effectiveness on the base LM, its performance on instruction-tuned LMs remains unexplored, limiting its contribution. For example, in the experiment for Figure 3, the performance with and without ALIGNEZ when the DPO dataset is expanded to 100% is not assessed.

3. Generalizability Issues: ALIGNEZ uses a statistical method (kNN) to obtain feature vectors for embedding editing during inference, raising concerns about its generalizability. It may only be effective when the inference input data is strongly correlated with the statistical data, making it difficult to handle out-of-distribution (OOD) situations.

4. Narrow Focus on Helpfulness: A significant limitation of the paper is its narrow focus on helpfulness, which does not convincingly demonstrate the overall effectiveness of ALIGNEZ. This raises doubts about whether ALIGNEZ is universally applicable to other aspects, such as safety or more complex alignment scenarios that involve a mix of helpfulness and safety. More extensive testing would strengthen the paper's claims.

5. Dependency on LLM's Instruction-Following Ability: Since ALIGNEZ relies on self-generated data to identify subspaces in the LM's embedding spaces corresponding to helpful and harmful directions for alignment, it requires a high degree of instruction-following capability from the LLM. This suggests that the method is better suited for models that have undergone instruction fine-tuning or alignment. However, the experiments are primarily conducted on the base pretrained model, which may not adequately reflect the method's potential effectiveness in more refined contexts.

**Questions:**

1. In the claim, 'With this nearly cost-free procedure, we effectively narrow the performance gap between pretrained and aligned models by 29.1% across two model architectures and five datasets,' how was the 29.1% calculated?

2. Line 239 may be intended to refer to $z_{help}$ and $z_{harm}$.

3. In Figure 4 (right), what insights can be drawn from the statement 'The lowest cosine similarities are observed in the middle layers (layers 10 to 25)'?

4. In line 303, the term "DPOed" is used, which is not a formal expression.

5. The main experiments lack detailed descriptions. It's not clear how to implement the method across different datasets.

---

> ### Author Response · Authors · 2024-11-20
> **Response to Reviewer yARA**
>
> Thank you for noting the **scalability, efficacy, and data efficiency of our approach**!
>
> - **On prompts used for self-generated data (W1).** We provide prompt details in Appendix A.4.1 and reference these in Section 2.1 (line 139) of our paper. For all experiments presented in the main paper, we consistently use this same set of prompts to generate synthetic data.
>
> - **On Figures 3 and 4 (W1).**
>     - For Figure 3: As described in Section 4.2 of the main paper, we use the default training split of UltraFeedback-binarized [1, 2] to train the DPO model and evaluate it on the default test split (long-form generation slice). To address the reviewer's concern, we will add this information in the Figure 3 caption in the corrected version of our manuscript.
>     - For Figure 4: Thank you for pointing this out! The figure reports the average performance across the five datasets listed in Table 1. We will update our manuscript to include this clarification.
>
> - **On the dependence on the base model (W2).** As stated in the Introduction section of our manuscript, **AlignEZ is specifically designed to leverage alignment signals from noisy, self-generated data produced by the base model**. It does this by selectively using preference data from points identified as similar in the latent space. The core intuition is that points within a localized region of the embedding space tend to exhibit similar properties [3]. By focusing only on preference data from these similar points, AlignEZ avoids noise from unrelated data and ensures the use of only relevant and important characteristics for alignment.
>
>     Existing instruction-tuned models are often trained on synthetic, noisy instruction pairs [4, 5], meaning the data they generate may also contain noise. Moreover, prior work has highlighted the risks of model collapse when synthetic data is overused [5-10]. AlignEZ addresses this issue with a test sample-specific approach that targets alignment by selecting only the most relevant synthetic data points. This strategy not only reduces the risks associated with synthetic data overuse but also ensures that alignment remains effective, even in challenging, data-constrained environments.
>
> - **On OOD scenarios (W3).** The purpose of AlignEZ is to be an inexpensive replacement for alignment techniques. **Addressing OOD concerns for alignment techniques is an orthogonal problem that applies to standard alignment methods as well as ours**. In fact, it is still very much an open problem, even for well-established approaches like PPO-based methods (e.g., RLHF) and DPO [11, 12]. Despite their effectiveness in many settings, these methods struggle to generalize to OOD scenarios.
>
> - **On helpfulness and safety aspects (W4).** As suggested by the reviewer, we compare AlignEZ with the baselines on 100 random samples of HH-RLHF; which consist both helpfulness and safety aspects. Similarly to the setup in the main paper, we use ground truth data for CAA and ITI, and use synthetic data for AlignEZ. We report the average Net Win ($\Delta \%$) across 3 random seeds:
>
>     |Model|Ours|ITI|CAA
>     |-|-|-|-|
>     Llama3.1| **7.67%** | -2% | 0.6%
>     Mistral3 |**14%** | 2.33% | 6.67%
>
> The result above shows the effectiveness of AlignEZ on alignment tasks with multiple dimensions (safety and helpfulness).
>
> - **On dependence on LLM's instruction-following ability (W5).** Our findings indicate that **AlignEZ works on base models---and does not require instruction-tuned models**, as shown by the significant alignment gain we show in Table 1 of out manuscript.
>
> - **On how the improvement average is calculated.** This is calculated by taking an average of all the Relative Improvement (RI) numbers (right-most column of Table 1).
>
> -  **On implementation.** As stated in the previous response, the prompts used for synthetic data generation are provided in Appendix A.4.1. We also provided our code as a .py file in the supplementary material provided in the initial submission.

---

> > ### Author Response · Authors · 2024-11-20
> > **Response to Reviewer yARA (cont.)**
> >
> > [1] Cui, G., Yuan, L., Ding, N., Yao, G., Zhu, W., Ni, Y., ... & Sun, M. (2023). Ultrafeedback: Boosting language models with high-quality feedback. arXiv preprint arXiv:2310.01377.
> >
> > [2] Tunstall, Lewis, Edward Beeching, Nathan Lambert, Nazneen Rajani, Kashif Rasul, Younes Belkada, Shengyi Huang et al. "Zephyr: Direct distillation of lm alignment." arXiv preprint arXiv:2310.16944 (2023).
> >
> > [3] Chen, M. F., Fu, D. Y., Adila, D., Zhang, M., Sala, F., Fatahalian, K., & Ré, C. (2022, August). Shoring up the foundations: Fusing model embeddings and weak supervision. In Uncertainty in Artificial Intelligence (pp. 357-367). PMLR.
> >
> > [4] Zhang, S., Dong, L., Li, X., Zhang, S., Sun, X., Wang, S., ... & Wang, G. (2023). Instruction tuning for large language models: A survey. arXiv preprint arXiv:2308.10792.
> >
> > [5] Albalak, A., Elazar, Y., Xie, S. M., Longpre, S., Lambert, N., Wang, X., ... & Wang, W. Y. (2024). A survey on data selection for language models. arXiv preprint arXiv:2402.16827.
> >
> > [6] Kazdan, J., Schaeffer, R., Dey, A., Gerstgrasser, M., Rafailov, R., Donoho, D. L., & Koyejo, S. (2024). Collapse or Thrive? Perils and Promises of Synthetic Data in a Self-Generating World. arXiv preprint arXiv:2410.16713.
> >
> > [7] Gerstgrasser, M., Schaeffer, R., Dey, A., Rafailov, R., Sleight, H., Hughes, J., ... & Koyejo, S. (2024). Is model collapse inevitable? breaking the curse of recursion by accumulating real and synthetic data. arXiv preprint arXiv:2404.01413.
> >
> > [8] Taori, R., & Hashimoto, T. (2023, July). Data feedback loops: Model-driven amplification of dataset biases. In International Conference on Machine Learning (pp. 33883-33920). PMLR.
> >
> > [9] Veprikov, A., Afanasiev, A., & Khritankov, A. (2024). A Mathematical Model of the Hidden Feedback Loop Effect in Machine Learning Systems. arXiv preprint arXiv:2405.02726.
> >
> > [10] Seddik, M. E. A., Chen, S. W., Hayou, S., Youssef, P., & Debbah, M. (2024). How bad is training on synthetic data? a statistical analysis of language model collapse. arXiv preprint arXiv:2404.05090.
> >
> > [11] Ahrabian, K., Lin, X., Patra, B., Chaudhary, V., Benhaim, A., Pujara, J., & Song, X. (2024). The Hitchhiker's Guide to Human Alignment with* PO. arXiv preprint arXiv:2407.15229.
> >
> > [12] Xu, S., Fu, W., Gao, J., Ye, W., Liu, W., Mei, Z., ... & Wu, Y. (2024). Is dpo superior to ppo for llm alignment? a comprehensive study. arXiv preprint arXiv:2404.10719.

---

> ### Comment · Reviewer_yARA · 2024-11-25
>
> Thanks for the clarifications. The authors have addressed my concerns. I have changed my rating accordingly.

---

### Official Review · Reviewer_Wxrr · 2024-10-31

**Soundness:** 3
**Presentation:** 3
**Contribution:** 3
**Rating:** 6
**Confidence:** 4

**Summary:**

The paper introduces AlignEZ, a cost-efficient approach for aligning pretrained language models without the need for large-scale ground-truth preference data or extensive computational resources. Instead, AlignEZ utilizes self-generated preference data and representation editing to adjust model outputs during inference. By modifying model representations to suppress undesirable traits and enhance preferred ones using identified subspaces, AlignEZ significantly improves model alignment. Experimental results across five datasets and two architectures demonstrate a 29.1% average improvement, narrowing the gap between pretrained and fine-tuned models. Additionally, AlignEZ shows potential for expediting more expensive alignment methods by enhancing models trained with limited ground-truth preference data.

**Strengths:**

1. The paper is well-written and easy to follow.
2. The performance results presented, particularly in Table 1, demonstrate the effectiveness of AlignEZ. As an inference-time alignment technique, AlignEZ significantly improves the alignment recovery compared to other test-time alignment baselines.
3. The authors conduct additional experiments to confirm that their method is orthogonal to prompting techniques such as URIAL. This enhances the practicality and robustness of the proposed approach.

**Weaknesses:**

1. The comparison with the baselines appears somewhat unfair. While the baseline test-time alignment techniques utilize ground-truth preference signals, these signals are sourced from a different dataset (HH-RLHF), introducing a distribution shift compared to the data used in AlignEZ. To provide a more balanced comparison, I suggest evaluating AlignEZ using the same HH-RLHF data during inference.

2. Concerning practicality and efficiency, the authors do not provide details on the overhead induced by their method, particularly in terms of inference latency or memory utilization. It would be beneficial to compare these overheads against those of the baseline methods.

3. It remains unclear whether AlignEZ affects other critical aspects, such as model safety or propensity for hallucination. I recommend including a discussion or experiments that assess the potential impact of AlignEZ on these factors.

4. A highly viable and practical application of test-time alignment is personalization, rather than just general helpfulness (which is already addressed by the aligned versions of many models). Moreover, the dataset used in this paper is somewhat outdated. Providing experiments or insights focused on personalization could offer valuable contributions and make the paper more relevant.

**Questions:**

1. What would the impact be if only the “helpfulness enhancement” or solely the “harmfulness suppression” (as described in Section 2.3) were applied individually? Could you provide insights or experiments on this?

2. Could using k-nearest neighbor (kNN) search to retrieve responses as in-context examples when prompting the model further enhance its alignment abilities, instead of just using this in the sample-conditional estimation of helpful and harmful direction.

---

> ### Author Response · Authors · 2024-11-20
> **Response to Reviewer Wxrr**
>
> Thank you for highlighting the **effectiveness and practicality of our test-time alignment method**!
>
> - **On comparison with baselines.** Thank you for pointing this out! As suggested by the reviewer, we compare AlignEZ with the baselines on 100 random samples of HH-RLHF. Similarly as the setup in the main paper, we use ground truth data for CAA and ITI, and use synthetic data for AlignEZ. We report the average Net Win ($\Delta \%$) across 3 random seeds:
>
> |Model|Ours|ITI|CAA
> |-|-|-|-|
> Llama3.1| **7.67%** | -2% | 0.6%
> Mistral3 |**14%** | 2.33% | 6.67%
>
> The table above shows that AlignEZ's **performance gain over the baseline persists even when the baselines use ground truth data form the same distribution as the test samples.**
>
> - **On inference latency and memory utilization.** Thank you for bringing up this point! We provide overhead comparison with baselines as follows:
>
> **Comparison with DPO (low data regime)**
>
> For DPO, we measured the total time required for both training and inference. For AlignEZ, we measured the end-to-end time, including synthetic data generation, alignment direction identification, and inference. Our goal was to equalize the amount of time taken in order to fairly compare performance across these methods. We obtained
>
> |Number of samples|DPO wall-time|AlignEZ wall-time|
> |-|-|-|
> 100 | **682s**  | 706s |
> 200 | 1372s | **1342s** |
> 300 | 1968s | **1893s** |
>
> The results highlight that indeed **AlignEZ with latency comparable to DPO in the low-data regime  delivers significantly better performance** in these scenarios, as shown in Figure 1 of the main paper. We note that our setup is actually highly favorable to DPO, as it **does not factor in the additional cost of obtaining the ground-truth data required by DPO**. If were were to factor this in, AlignEZ would achieve further superiority.
>
> **Comparison with embedding editing baselines**
> **AlignEZ incurs similar overhead as baseline representation editing methods** (CAA and ITI), detailed as follows:
> 1. Embedding modification:
>     - CAA and ITI modify embeddings by performing scalar multiplication followed by the addition of a fixed vector to the model's activations.
>     - AlignEZ modifies embeddings by performing a dot product followed by two additions (removing harmful vectors and adding helpful vectors).
>     - CAA, ITI, and AlignEZ's latency is $O(d)$ where $d$ is the embedding dimension.
>
> 2. Alignment direction identification:
>     - CAA uses PCA for this step, while AlignEZ uses SVD, resulting in comparable latency costs.
>     - ITI, however, trains $H$ classifiers ($H$ is the number of attention heads), which incurs the highest latency cost among the methods for this step.
>
> 3. AlignEZ has an extra step to generate synthetic data and find the nearest points for each test sample using kNN (done only once in the beginning). kNN incurs $O(nd)$ cost, with $n$ the test sample size and $d$ the embedding dimension. For synthetic data generation, with fast inference library such as vLLM (https://github.com/vllm-project/vllm), generating data for 100 samples only takes 30 seconds on an A100 GPU. It is worth noting that we did not account for the cost of obtaining ground-truth data required by ITI and CAA---**but we do for AlignEZ, suggesting that when accounting for data collection complexity, AlignEZ would have an even better relative performance**.
>
> - **On effect on safety and hallucination.** As suggested by the reviewer, we perform an experiment to test AlignEZ's impact on safety and hallucination.
>
>     **Safety.** We tested AlignEZ on two safety datasets, namely MaliciousInstruct [1] and JailBreakBench [2], and report the Net Win ($\Delta \%$) below:
>
>     |Model|MaliciousInstruct|JailbreakBench
>     |-|-|-|
>     Llama3.1| 3% | 6% |
>     Mistral3 | 1% | -3% |
>
>     The results show that **AlignEZ provides a modest safety improvement for Llama 3.1 and has minimal impact on safety for Mistral 3**. This indicates that AlignEZ does not negatively affect safety and may even present opportunities for developing specialized versions tailored for safety-critical applications.
>
>     **Hallucination.** We conducted the FActScore test [3], an evaluation method for assessing the degree of hallucination in LLM-generated responses. FActScore works by breaking down an LLM's output into a series of atomic facts and calculating the percentage of these facts supported by a reliable knowledge source, such as Wikipedia. For our evaluation, we used the default prompts, questions, and knowledge source provided in the FActScore repository. The scores range from 0 to 1, where a **higher score indicates a less hallucinated response**.
>
>     |Model|Base Model|Base Model + AlignEZ
>     |-|-|-|
>     Llama3.1| 0.444 | 0.436 |
>     Mistral3 | 0.458| 0.452 |
>
>     The results show that **AlignEZ has little to no effect to the original model's degree of hallucination**, maintaining its factual accuracy.

---

> > ### Author Response · Authors · 2024-11-20
> > **Response to Reviewer Wxrr (cont.)**
> >
> > - **On applications to personalization.** Thank you for the suggestion! Following the reviewer's advice, we tested AlignEZ on personalization tasks using the LaMP benchmark [4]. Specifically, we evaluated on:
> >
> >     - LaMP 2: Personalized movie tagging (classification task)
> >     - LaMP 7: Personalized tweet paraphrasing (open-ended generation task)
> >
> >     We ran AlignEZ on the Mistral-7B-Instruct-v0.2 model and compared it with the following baselines: The instruct model without AlignEZ, LLM-REC (prompting-based) [5], ALOE (SFT-based) [6]. We used the default data splits and evaluated using the standard metrics from the benchmark:
> >     - LaMP 2: Accuracy and F-1
> >     - LaMP 7: ROUGE-1 (R-1) and ROUGE-L (R-L)
> >
> >     Consistent with our main paper experiments, we use self-generated preference data for AlignEZ, and use the ground-truth data for the baselines.
> >
> >     **LaMP 2**
> >
> >     |Method|Accuracy ($\uparrow$)|F1 ($\uparrow$)|
> >     |-|-|-|
> >     | Instruct Model | 0.198 | 0.236
> >     | LLM-REC| 0.262 | 0.309|
> >     | ALOE | 0.307 | 0.220 |
> >     | AlignEZ | **0.407** | **0.358**
> >
> >      **LaMP 7**
> >
> >     |Method|R-1 ($\uparrow$)|R-L ($\uparrow$)|
> >     |-|-|-|
> >     | Instruct Model | 0.354 | 0.295
> >     | LLM-REC| 0.183 | 0.144
> >     | ALOE | 0.362 | 0.313
> >     | AlignEZ | **0.398** | **0.349**
> >
> >     This result demonstrates **AlignEZ's effectiveness in aligning LLM to more specific preferences in personalization tasks -- notably even surpassing an SFT based baseline** (ALOE).
> >
> >
> > - **On isolating $\theta^{help}$ and $\theta^{harm}$.** Based on the reviewer's suggestion, we conducted an experiment to evaluate the individual effects of increasing $\theta^{help}$ and reducing $\theta^{harm}$. The Net Win ($\Delta \%$) for each case is reported below:
> >
> >     **Model: Mistral 3**
> >     |Dataset|Increase $\theta^{help}$|Reduce $\theta^{harm}$|Both|
> >     |-|-|-|-|
> >     oasst| -21% | 12% | 16% |
> >     MT| -6% | 3% | -1% |
> >     helpful-base| -25% | 13% | 12% |
> >     self-instruct| -14% | 0% | 11% |
> >     koala| -27% | 15% | 8% |
> >
> >     **Model: Llama 3.1**
> >     |Dataset|Increase $\theta^{help}$|Reduce $\theta^{harm}$|Both|
> >     |-|-|-|-|
> >     oasst| 7% | 0% | 7% |
> >     MT| -5% | 29% | 7% |
> >     helpful-base| -9% | 13%  | -1%  |
> >     self-instruct| -13% | 10%  | 16% |
> >     koala| 1% | -18% | 0% |
> >
> >     In most cases, reducing $\theta^{harm}$ gives the best performance. However, when reducing $\theta^{harm}$ alone does not lead to any improvement (e.g., Mistral 3 self-instruct, Llama 3.1 oasst and koala), combining both steps—reducing $\theta^{harm}$ followed by increasing $\theta^{help}$-- restores performance gain. This suggest that **both components are necessary for achieving optimal performance**.
> >
> >
> >
> > - **On in-context learning.** Thank you for the suggestion! We tested the idea of **using self-generated data as in-context learning examples and found that it degraded AlignEZ's performance**. One notable trend was that the model generated much shorter responses overall. We hypothesize that this is due to two factors:
> >     1. Feeding noisy self-generated data into the model likely propagates noise.
> >     2. **Using in-context examples consumes the LLM's already limited context window**, reducing the space available for processing the actual input.
> >
> >
> > [1] Huang, Y., Gupta, S., Xia, M., Li, K., & Chen, D. (2023). Catastrophic jailbreak of open-source llms via exploiting generation. arXiv preprint arXiv:2310.06987.
> >
> > [2] Chao, P., Debenedetti, E., Robey, A., Andriushchenko, M., Croce, F., Sehwag, V., ... & Wong, E. (2024). Jailbreakbench: An open robustness benchmark for jailbreaking large language models. arXiv preprint arXiv:2404.01318.
> >
> > [3] Min, S., Krishna, K., Lyu, X., Lewis, M., Yih, W. T., Koh, P. W., ... & Hajishirzi, H. (2023). Factscore: Fine-grained atomic evaluation of factual precision in long form text generation. arXiv preprint arXiv:2305.14251.
> >
> > [4] Salemi, A., Mysore, S., Bendersky, M., & Zamani, H. (2023). Lamp: When large language models meet personalization. arXiv preprint arXiv:2304.11406.
> >
> > [5] Lyu, Hanjia, Song Jiang, Hanqing Zeng, Yinglong Xia, Qifan Wang, Si Zhang, Ren Chen, Christopher Leung, Jiajie Tang, and Jiebo Luo. "Llm-rec: Personalized recommendation via prompting large language models." arXiv preprint arXiv:2307.15780 (2023).
> >
> > [6] Wu, S., Fung, M., Qian, C., Kim, J., Hakkani-Tur, D., & Ji, H. (2024). Aligning LLMs with Individual Preferences via Interaction. arXiv preprint arXiv:2410.03642.

---

> > > ### Comment · Reviewer_Wxrr · 2024-11-24
> > >
> > > Thanks for the comprehensive responses, I think most of my concerns have been clearly addressed. I am curious about the new experiement of isolating increasing helpfulness and reducing harmfulness, it looks like in most cases increasing helpfulness actually make it worse. Can you provide any comments or explanation for this?

---

> > > > ### Author Response · Authors · 2024-11-24
> > > >
> > > > Thank you for the response! Adding a single vector requires very carefully tuning its magnitude---if this is too large, the model's outputs change too much and performance degrades. This observation is consistent with prior work, such as [1, 2]. For example, the authors in [1] saw consistent performance degradation when the scaling constant for the editing vector is larger than a certain threshold. There is a similar finding in [2].
> > > >
> > > > The intuition is that using multiple vectors balances some of these effects, leading to less performance degradation. This also reduces the need to carefully select the scaling hyperparameter for the editing vector.
> > > >
> > > > [1] Li, K., Patel, O., Viégas, F., Pfister, H., & Wattenberg, M. (2024). Inference-time intervention: Eliciting truthful answers from a language model. Advances in Neural Information Processing Systems, 36.
> > > >
> > > > [2] Adila, D., Zhang, S., Han, B., & Wang, Y. (2024). Discovering Bias in Latent Space: An Unsupervised Debiasing Approach. arXiv preprint arXiv:2406.03631.

---

> > > > > ### Comment · Reviewer_Wxrr · 2024-11-25
> > > > >
> > > > > Thanks for the follow up response. I have raised my rating accordingly.

---

### Official Review · Reviewer_sGAG · 2024-11-04

**Soundness:** 3
**Presentation:** 3
**Contribution:** 3
**Rating:** 5
**Confidence:** 2

**Summary:**

This paper introduces AlignEZ, a method for aligning large language models (LMs) with human preferences without using additional training data or fine-tuning. The approach addresses the high costs associated with traditional alignment methods that require extensive human-annotated data and fine-tuning. AlignEZ leverages self-generated preference data by prompting the base model to produce examples of “helpful” and “harmful” responses, thereby creating synthetic data that approximates human preferences. During inference, AlignEZ performs representation editing, modifying model embeddings to accentuate desirable (helpful) and reduce undesirable (harmful) components. This enables alignment without training, relying instead on adjustments to the model’s representations in real time.

Empirically, AlignEZ narrows the performance gap between base and aligned models by 29.1% on average across multiple datasets and architectures, demonstrating that it can boost alignment quality in a cost-effective manner. In experiments, AlignEZ also expedites traditional alignment processes like Direct Preference Optimization (DPO), enhancing model performance even when only a small subset of ground-truth data is available. Additionally, AlignEZ integrates effectively with prompting techniques, yielding further improvements beyond what prompting alone can achieve.

**Strengths:**

1. By eliminating reliance on costly fine-tuning and human-annotated preference data, AlignEZ offers a more resource-efficient alternative to traditional alignment techniques, making it potentially scalable for larger or real-time applications.
2. AlignEZ presents a conceptually straightforward approach, leveraging self-generated preference data and representation editing to achieve alignment without the need for extensive fine-tuning or additional ground-truth annotations.
3.  Empirically, the approach demonstrates substantial performance gains over DPO baselines, showing that AlignEZ is effective even when traditional alignment data is limited. It also improves data efficiency by expediting existing RLHF methods, like DPO.
4. Moreover, AlignEZ integrates well with prompting strategies, enhancing alignment performance beyond what prompting alone achieves and expanding its usability with other alignment techniques. Prompting method is another cheap way to align LLMs without much cost and it's good to see both methods can be combined strongly.

**Weaknesses:**

1. While AlignEZ combines self-generated preference data and representation editing effectively, both of these approaches are well-established methods. Techniques for generating synthetic preference data have been extensively studied in self-alignment literature, which limits the technical novelty of this work. To strengthen its contribution, the authors could further emphasize unique aspects of their integration of these techniques or explore additional novel dimensions, such as adaptive mechanisms for more refined, context-dependent alignment adjustments.
2. The concept of "free" alignment in AlignEZ may be somewhat overstated, as both the self-generation of preference data and embedding modifications require non-negligible computational resources. While ALIGNEZ reduces reliance on human-labeled data, it does not entirely eliminate costs, especially when scaling to larger models. Clarifying these claims by discussing cost reductions relative to traditional methods, rather than "free" alignment, would provide a more balanced and realistic portrayal of AlignEZ's cost efficiency.
3. Similarly, the title, “Is Free Self-Alignment Possible?” may prioritize entertainment value over clarity, providing limited insight into the paper’s actual contributions. A more precise framing, such as “Cost-Efficient Self-Alignment through Representation Editing and Self-Generated Data,” could better communicate the scope and implications of the study.
4. The theoretical analysis presented in Section 3 is built on simplifying assumptions that may not fully capture the complexities of real-world model behavior. Specifically, the assumption that LLM space is orthonormal is oversimplified. Also, the conclusion is unclear enough to specify whether the singular vectors from AlignEZ are strong enough or whether the kNN smoothing is good enough, etc
5. The empirical results are based on relatively small language models (7B and 8B parameters), which, given the rapid advancements in AI, are now considered less representative of state-of-the-art capabilities. Applying ALIGNEZ to significantly larger models (e.g., 100B+ parameters) would provide stronger evidence of its scalability and effectiveness in larger, more complex architectures. While it is challenging to access larger models due to their proprietary nature, exploring ways to adapt ALIGNEZ for environments where parameter access is limited, or testing it on larger open-source models, could strengthen its empirical validation. Additionally, the reliance on open-source models might be seen as a minor limitation, as this restricts ALIGNEZ’s applicability to cases where model weights are accessible, potentially narrowing its practical impact in real-world, production-level systems that often employ closed-source models.

**Questions:**

1. While the paper shows the compatibility of this method with the prompting method, I wonder about the head-to-head comparison of the proposed technique and prompting technique in improving the alignment performance of LLMs, since both are very cheap methods,
2. Any fun visualization or analyses on the "help" and "harm" vector space?
3. Any thoughts on the reward-guided test-time alignment techniques? This is related to the efficiency alignment techniques, although it's not closely related to the techniques in this paper per se.

---

> ### Author Response · Authors · 2024-11-20
> **Response to Reviewer sGAG**
>
> Thank you for noting the **resource efficiency and strong performance demonstrated by our approach**!
>
> - **On contribution.** While synthetic data generation and representation editing are well-established methods, **their combination as well as their use in alignment has not, to our knowledge, been explored**. Our baseline results show that current SOTA representation engineering methods fail to improve alignment because alignment requires adapting to nuanced and loosely defined knowledge, unlike tasks such as improving truthfulness (e.g., ITI) or adopting linguistic styles (e.g., CAA).  **AlignEZ bridges this gap by introducing a novel strategy** that selectively uses synthetic data points identified as similar in the latent space,  enabling precise, test sample-specific alignment while filtering out noise from unrelated data points.
>
>
>     In Figure 1 of our manuscript, we demonstrate that **AlignEZ outperforms traditional alignment methods like DPO, particularly in low-data scenarios**. This reflects real-world conditions, especially when synthetic data is used—a setup that has been gaining traction recently [1, 2, 5]. While prior work highlights the risks of model collapse when synthetic data is overused [3, 4, 5], **AlignEZ tackles this challenge by introducing a test sample-specific approach** that targets alignment by selecting only the most relevant synthetic data points. This strategy not only mitigates the risks of model collapse in synthetic data usage but also ensures effective alignment in challenging, data-constrained conditions.
>
> - **On 'free' claim and title.** Thank you for the thoughtful suggestions! We agree with the reviewer that our method does incur a non-zero cost. However, we want to point out that this cost is significantly lower compared to traditional alignment methods (e.g., RLHF and DPO). Unlike DPO, we do not need to spend any time or cost acquiring data, and we do not run any SGD iterations for fine-tuning.
>
>     The primary cost associated with AlignEZ arises from generating synthetic data, which is relatively inexpensive. Hosting the model locally eliminates API call expenses, and inference speed can be further improved using tools like vLLM (https://github.com/vllm-project/vllm). To address the concern about terminology, we are happy to revise the title to use "inexpensive" instead of "free."
>
> - **On theoretical analysis.**
> The assumption that LLM representations can be decomposed into latent concepts is widely adopted and has been supported by prior work across diverse contexts, e.g., [8-13]. These studies validate the practical utility of such assumptions in analyzing and leveraging model behavior. While we adopt the orthonormality assumption for clarity in explanation and derivation, **our method is not reliant on it**. Specifically, if the concept vectors are not orthonormal, the analysis can proceed by representing them under a change of basis, ensuring the results remain valid. Regarding the conclusion of the theorem, we clarify that it is encapsulated by the condition $\sigma_{\text{linguistic}}=C \cfrac{\max_{q \in \text{k-NN}(q_x)}d(q_x, q)}{\sqrt{k}}$. This implies that increasing k can be effective if all of k-NN examples are sufficiently close to the query, leading to a decrease in the term. This theoretical insight aligns with empirical results, as demonstrated in Figure 4(a).
>
>
> - **On evaluation on larger models.** We would like to point out that applicability to proprietary models is not a limitation specific to AlignEZ. In fact, **all alignment methods require access to model weights to implement** alignment effectively. AlignEZ imposes no additional requirements beyond this standard access.
>
>     To address the reviewer's suggestion, we have evaluated AlignEZ on a larger open-source model, Llama 3.1 70B, to further demonstrate its applicability and scalability. On the oasst dataset, AlignEZ provides a Net Win ($\Delta \%$) of 4\%, **illustrating that the performance gain persist for larger models**.

---

> ### Author Response · Authors · 2024-11-20
> **Response to Reviewer sGAG (cont.)**
>
> - **On head to head comparison with prompting technique.** As suggested by the reviewer, we conducted a head-to-head comparison between URIAL [6], a prompting-based method, and AlignEZ. Since URIAL was specifically optimized for the just-eval dataset [6] used in our main paper, we ensured a fair comparison by evaluating both methods on 100 randomly selected samples from HH-RLHF, with results averaged across three random seeds. We report the Net Win ($\Delta \%$) = Win\%-Lose\%  for AlignEZ below
>
> |Model|AlignEZ Net Win ($\Delta \%$)|
> |-|-|
> Llama3.1| 14% |
> Mistral 3 | 12.67% |
>
> **The positive Net Win scores highlight AlignEZ's effectiveness and superiority compared to URIAL**.  Prompting methods like URIAL are compute-efficient and compatible with proprietary models but come with notable drawbacks, such as a reliance on significant human effort to craft optimized prompts [6]. These methods also incur additional costs from context window usage and extra tokens. AlignEZ addresses these challenges through representation engineering, which eliminates the need for context window overhead and enables the use of generic prompts (as detailed in Appendix A.4.1 of our manuscript). We have added this result in Appendix E in our updated manuscript.
>
> - **On $\theta^{help}$ and $\theta^{harm}$.** Thank you for your suggestion! We have added $\theta^{help}$ and $\theta^{harm}$ visualizations as Appendix D in our updated manuscript. The visualizations shows that $\theta^{help}$ and $\theta^{harm}$ form distinct and separable clusters even in this low-dimensional representation (2 dimension PCA).
>
> - **On reward-model test-time alignment methods.** While reward-model-guided methods are compute-efficient at test time, we argue that they incur higher costs during training. **Training a reward model requires access to human-annotated preference data and fine-tuning compute**, making it computationally prohibitive when alignment must adapt to changing or evolving preferences.
>
>     Additionally, we argue that AlignEZ offers greater controllability and interpretability compared to reward-model-based methods. With AlignEZ, users can directly prompt for synthetic data (as detailed in Appendix A.4.1), allowing for easy customization. In contrast, reward-model-based methods require access to the reward model's training data to enable any degree of interpretability or control [7].
>
> [1] Kazdan, J., Schaeffer, R., Dey, A., Gerstgrasser, M., Rafailov, R., Donoho, D. L., & Koyejo, S. (2024). Collapse or Thrive? Perils and Promises of Synthetic Data in a Self-Generating World. arXiv preprint arXiv:2410.16713.
>
> [2] Gerstgrasser, M., Schaeffer, R., Dey, A., Rafailov, R., Sleight, H., Hughes, J., ... & Koyejo, S. (2024). Is model collapse inevitable? breaking the curse of recursion by accumulating real and synthetic data. arXiv preprint arXiv:2404.01413.
>
> [3] Taori, R., & Hashimoto, T. (2023, July). Data feedback loops: Model-driven amplification of dataset biases. In International Conference on Machine Learning (pp. 33883-33920). PMLR.
>
> [4] Veprikov, A., Afanasiev, A., & Khritankov, A. (2024). A Mathematical Model of the Hidden Feedback Loop Effect in Machine Learning Systems. arXiv preprint arXiv:2405.02726.
>
> [5] Seddik, M. E. A., Chen, S. W., Hayou, S., Youssef, P., & Debbah, M. (2024). How bad is training on synthetic data? a statistical analysis of language model collapse. arXiv preprint arXiv:2404.05090.
>
> [6] Lin, B. Y., Ravichander, A., Lu, X., Dziri, N., Sclar, M., Chandu, K., ... & Choi, Y. (2023, December). The unlocking spell on base llms: Rethinking alignment via in-context learning. In The Twelfth International Conference on Learning Representations.
>
> [7] Carroll, M., Foote, D., Siththaranjan, A., Russell, S., & Dragan, A. AI alignment with changing and influenceable reward functions. 2024. URl: https://arxiv.org/abs/2405.17713.
>
> [8] Dev, Sunipa, et al. "OSCaR: Orthogonal Subspace Correction and Rectification of Biases in Word Embeddings." EMNLP 2021.
>
> [9] Dalvi, Fahim, et al. "Discovering Latent Concepts Learned in BERT." ICLR 2022.
>
> [10] Trager, Matthew, et al. "Linear spaces of meanings: compositional structures in vision-language models." CVPR 2023.
>
> [11] Chuang, Ching-Yao, et al. "Debiasing vision-language models via biased prompts." arXiv 2023.
>
> [12] Park, Kiho, Yo Joong Choe, and Victor Veitch. "The Linear Representation Hypothesis and the Geometry of Large Language Models." ICML 2024.
>
> [13] Jiang, Yibo, Bryon Aragam, and Victor Veitch. "Uncovering Meanings of Embeddings via Partial Orthogonality." NeurIPS 2023.

---

> ### Author Response · Authors · 2024-11-25
>
> Dear Reviewer,
>
> We thank you again for your feedback, questions, and suggestions! We believe we have answered all of your questions in our responses and the updated draft. If you have additional questions, we would love to answer them!
>
> The Authors

---

> ### Author Response · Authors · 2024-11-28
>
> Dear Reviewer,
>
> Thank you again for your feedback and insightful questions. During the rebuttal period, we revisited your suggestions regarding the potential for adding more novelty axes, such as adaptive mechanisms for refined, context-dependent alignment adjustments (W1).
>
> We focus on the latter idea, as AlignEZ incorporates a context-dependent mechanism. The basic idea for context-dependent alignment is demonstrated in our two-stage prompting process, as detailed in Section 2.1 and illustrated in Figure 2 of the manuscript.
>
> 1. Stage 1: The LLM is prompted to identify characteristics of helpful and unhelpful responses **tailored to the specific context-dependent** test query. This step dynamically adapts to the query, ensuring that the derived characteristics are contextually relevant.
> 2. Stage 2: Using the characteristics generated in Stage 1, the LLM is prompted to produce a response to the test query. This results in preference samples that are explicitly aligned to the context of each question.
>
> To test the potential of AlignEZ for context-dependent alignment, we conducted **additional experiments** on the just-eval-instruct dataset [6], splitting the evaluation by task/context type. Below, we present the Net Win (Δ) results for these task-specific splits.
>
> |Model|Coding+Math|Reasoning
> |-|-|-|
> Llama3.1| 10\% | 8\% |
> Mistral3 | 8\% | 0\%  |
>
> These results demonstrate that **AlignEZ can effectively perform context-dependent alignment, achieving Net Win (Δ) improvements for several task-specific and challenging scenarios like Coding and Math**. In the case of Llama 3.1, the mechanism also enhances performance on reasoning tasks.
>
> We appreciate the suggestion and hope this clarifies and strengthens our response.

---

> > ### Comment · Reviewer_sGAG · 2024-11-28
> > **Thanks for the great rebuttal**
> >
> > I thank the authors for the comprehensive rebuttal effort, and I appreciate the new results. My major concern is still centred around the methodological novelty, which is kind of incremental, IMHO. There are lots of cost-efficient alignment methods now, from prompting, to representation engineering, to reward-based decoding, etc. This paper needs to better find its position in this complex landscape; it definitely is not the ultimate solution to this research direction.
> >
> > I can raise the score but considering the mediocre technical novelty and comprehensive empirical results with some theoretical analysis, I believe this paper is in the boradeline and it's up to ACs to make decisions whether to accept this or not.

---

> > > ### Author Response · Authors · 2024-11-28
> > >
> > > We appreciate your reply and are happy to clarify further. For alignment, the methodological space is the product of data and algorithms used. We detail these choices and where AlignEZ fits in:
> > >
> > > - Data: The set of choices for alignment data are (i) real data, (ii) combinations of real and synthetic data, (iii), synthetic data produced by a more powerful pretrained model, (iv) synthetic data produced by the base model to be aligned. These are ordered from least efficient/highest requirements to most efficient/lowest requirements.
> > >
> > > - Algorithm family: The set of choices are given by (a) token-space methods (i.e., modifications to decoding), (b) weight space methods (i.e., training, fine-tuning), (c) representation space methods, and (d) prompting. Again, here (c) and (d) are most efficient, while (a) and (b) are the most expensive. In particular (a) requires access to a high-quality reward model, as we detail below.
> > >
> > > **AlignEZ's location**: The goal is to use the most efficient combination of data and algorithm. Our approach uses fully synthetic data produced by the model itself (iv) with representation engineering (c). As we explain below, this is essentially the most efficient general choice possible.
> > >
> > > We observe as well that the description above (the breakdown of these methods) forms another contribution of our work. It is reflected in Sections 1, 2, 5.
> > >
> > > **AlignEZ methodical novelty**: As demonstrated in our evaluation, the combination of these two efficient choices for data and algorithm (synthetic data and representation editing/engineering) **is not straightforward**. In addition to the novelty of using the combination, we introduce innovations that enable it to produce high-quality alignment results. We do this by (1) generating preference samples *specific to the test query* with our two-step querying approach, (2) performing per-sample editing for each test query, by using only preference samples from queries relevant to it (identified by closeness in the embedding space).
> > >
> > > More details on algorithms and the basic motivation for our design in AlignEZ:
> > >
> > > - (d) Prompting: While efficient, prompting requires significant human effort to design optimal prompts for each task and these are usually **not transferable** across models and tasks. These flaws motivated us to use the next most efficient method, representation engineering. Additionally, prompting uses the model's context window, which can increase latency. In contrast, our method leverages a single generic prompt to extract the model's insights for subsequent representation engineering steps, eliminating the need for manual prompt optimization. Conveniently, AlignEZ is fully compatible with any prompt-based method.
> > >
> > > - (c) Representation editing and engineering: AlignEZ **uses a representation editing component** (and fits into this family of methods). However, using off-the-shelf representation engineering methods (without AlignEZ's innovations) is insufficient: we compared our approach with such methods and found that they often underperform in alignment tasks. This is because alignment requires adding nuanced, less well-defined knowledge compared to the structured tasks these methods were originally designed for. Our method addresses this gap by performing the representation editing specific to each query points, using only synthetic data generated by other queries closest to it in the latent space
> > >
> > > - (a) Reward-Based Decoding: Although reward-based decoding is cost-effective at test time, it incurs significant up front costs for training reward models, such as collecting training data and conducting the training itself. This makes it impractical for dynamic human preferences, which can evolve over time. Our method, AlignEZ, avoids these requirements. This makes it a lightweight and adaptable solution for aligning LLMs to new preference sets.
> > >
> > > [1] Carroll, M., Foote, D., Siththaranjan, A., Russell, S., & Dragan, A. AI Alignment with Changing and Influenceable Reward Functions. 2024. URL: https://arxiv.org/abs/2405.17713.

---

### Author Response · Authors · 2024-11-20
**General response to all reviewers**

We thank the reviewers for their thoughtful feedback and questions. Before proceeding with in-depth responses, we would like to highlight some benefits of our work noted by the reviewers:
- Our method is **data efficient and scalable** (reviewers sGAG and yARA).
- We demonstrate **strong alignment gain** (reviewers sGAG, Wxrr and yARA).
- Our method is **practical and can be easily integrated with other cheap alignment methods** (reviewers sGAG and Wxrr).

We respond to two common questions.
- **On multiple alignment axes** (Wxrr and yARA). We test AlignEZ on 100 random samples of HH-RLHF; which consist of **both helpfulness and safety aspects**. In a similar way to the setup in the main paper, we use ground truth data for CAA and ITI, and use synthetic data for AlignEZ. We report the average Net Win ($\Delta \%$) across 3 random seeds:

    |Model|Ours|ITI|CAA
    |-|-|-|-|
    Llama3.1| **7.67%** | -2% | 0.6%
    Mistral3 |**14%** | 2.33% | 6.67%

    These results demonstrate that on **datasets featuring multiple alignment axes—such as helpfulness and safety—AlignEZ achieves clear alignment gains**. This highlights its robustness and adaptability in addressing multiple dimensions of alignment.

- **On contribution and application to other tasks** (sGAG and Wxrr). AlignEZ is designed to provide alignment gains ***under limited data and compute resources by combining synthetic data with representation editing. This combination is non-trivial***, as baseline results demonstrate that existing SOTA representation engineering methods fail to improve alignment. Furthermore, synthetic data is inherently noisier than human-annotated data, posing additional challenges. AlignEZ overcomes this by harnessing alignment signal from the noisy synthetic data, focusing only on points identified as similar in the latent space. This targeted approach ensures more effective and precise alignment for each test sample. In Figure 1 of our main manuscript, we demonstrate that indeed **AlignEZ outperforms traditional alignment methods like DPO in low-data scenarios.**

    One real-life application where data and compute is inherently restricted are  personalization tasks. It is computationally prohibitive to perform fine-tuning for each individual user, and user-specific preference data is naturally limited. As suggested by reviewer Wxrr, we apply AlignEZ on the LaMP personalization benchmark [1], specifically on personalized movie tagging (LaMP 2) and personalized tweet paraphrasing (LaMP 7) tasks and use the benchmark default metrics for evaluation. We perform AlignEZ on the Mistral instruct model, and compare it with the vanilla instruct model, LLM-REC (prompting-based) and ALOE (SFT-based). We observe the following results:

    **LaMP 2**

    |Method|Accuracy ($\uparrow$)|F1 ($\uparrow$)|
    |-|-|-|
    | Instruct Model | 0.198 | 0.236
    | LLM-REC| 0.262 | 0.309|
    | ALOE | 0.307 | 0.220 |
    | AlignEZ | **0.407** | **0.358**

     **LaMP 7**

    |Method|R-1 ($\uparrow$)|R-L ($\uparrow$)|
    |-|-|-|
    | Instruct Model | 0.354 | 0.295
    | LLM-REC| 0.183 | 0.144
    | ALOE | 0.362 | 0.313
    | AlignEZ | **0.398** | **0.349**

    These results showcase **AlignEZ’s ability to achieve effective alignment even in resource-constrained personalization tasks**.

[1] Salemi, A., Mysore, S., Bendersky, M., & Zamani, H. (2023). Lamp: When large language models meet personalization. arXiv preprint arXiv:2304.11406.

---

### Author Response · Authors · 2024-12-03
**Discussion Summary**

Dear Reviewers/AC/SAC/PC,

We appreciate your feedback and the discussion with you. We summarize the following takeaways:

- **Additional experiments.** As suggested by Reviewer Wxrr, we evaluated the impact of AlignEZ on hallucination and safety. AlignEZ achieves significant alignment improvements without any effect on the model's original level of hallucination. It also has a moderate impact on safety in Llama3.1. Details are provided in Appendix E.

- **Reframing contributions.** The discussion with Reviewer sGAG helped reframe our work's contributions: out of a large design space, AlignEZ enables using the combination of the two most efficient data and algorithm choices for general alignment: synthetic data and representation engineering. This required non-trivial innovations: (1) generating preference samples specific to the test query with our two-step querying approach, (2) performing per-sample editing for each test query by using only preference samples from queries relevant to it (identified by closeness in the embedding space).

---

### Meta-Review · Area_Chair_eQpb · 2024-12-14

**Metareview:**

ALIGNEZ is a cost-effective approach for aligning language models through self-generated preferences and representation editing, eliminating the need for additional training. However, reviewers have raised concerns about its limited innovation and a lack of clarity in the methodology. Furthermore, the paper falls on the borderline regarding quality and contribution. I recommend rejection.

**Additional Comments On Reviewer Discussion:**

Although the authors provided additional results, Reviewer sGAG remains concerned that the method is incremental and lacks novelty. As he/she suggested, I agree that this paper falls into the borderline one.

Reviewer Wxrr raised concerns about the unfair comparison with baselines, the practicality and efficiency of the proposed method, and its applicability to other personalities.


While the authors made efforts to address reviewers' confusion and **even reframed the contribution**, these revisions remain insufficient. Considering these points, I recommend rejection.

---

### Decision · Program_Chairs · 2025-01-22

Reject